# Hidden No More: Attacking and Defending Private Third-Party LLM Inference

Rahul Thomas [1 2]  Louai Zahran [1]  Erica Choi [1 3]  Akilesh Potti [1]  Micah Goldblum [1 3]  Arka Pal [1]

## Abstract

Recent advances in Large Language Models (LLMs) have led to widespread adoption of third-party inference services, raising critical privacy concerns. In this work, we introduce a novel reconstruction technique that can recover original prompts from hidden states with nearly perfect accuracy across multiple state-of-the-art LLMs in the increasingly important open-weights setting. Although the attack is conceptually simple, it has not – to the best of our knowledge – previously been described nor shown to work practically. Furthermore, our attack remains effective against various permutation and noise-based defenses, challenging assumptions about the security of previously proposed schemes. To address these vulnerabilities, we propose Cascade, a multi-party inference scheme that leverages sharding in the sequence dimension to retain privacy of the user input. Through theoretical analysis and empirical evaluation, we demonstrate that Cascade is secure against both our attack as well as previous methods, while maintaining computational and communication efficiency. Our findings highlight the importance of rigorous security analysis in privacy-preserving LLM inference and offer practical solutions for secure deployment.

## 1. Introduction

Modern large language models (LLMs) now often comprise hundreds of billions of parameters, necessitating significant hardware resources for deploying them for inference. In particular, recent *open-weights* models demonstrate cutting-edge performance (DeepSeek-AI et al., 2025; Qwen et al., 2025), but remain difficult for many to run. Individuals and organizations have therefore begun to increasingly rely on third-party LLM inference services that host these models. This raises significant privacy implications, particularly in domains where confidentiality of data is paramount, such as healthcare, finance and legal applications, and in jurisdictions where data privacy is subject to regulations (e.g. GDPR in Europe). As such, a growing area of research interest is the creation of inference methodologies and schemes that protect the privacy of user prompts.

One approach to privacy-preserving-inference is to have multiple parties jointly perform the inference, with the idea that each party cannot reconstruct the input with the information that it is given in the protocol. This approach is known as Secure Multi-Party Computation (SMPC) and has a long history of application to general functions (Yao, 1982; Goldreich et al., 1987). Recently, the methodologies of SMPC have been applied to LLMs (Huang et al., 2022; Hao et al., 2022; Pang et al., 2023; Akimoto et al., 2023; Dong et al., 2023b; Li et al., 2024). However, SMPC methods introduce significant computational and communication overhead, particularly so at non-linearities in the model.

Thus, other works aim to mitigate the punitive costs of SMPC through statistical obfuscation. Recent work (Zheng et al., 2024; Yuan et al., 2024; Luo et al., 2024) has used the permutation-equivariance of transformers (Xu et al., 2024) to propose permutation-based schemes for private inference. In these schemes, hidden states are revealed as permuted plaintext to the party performing inference. To justify security, these works refer to the extremely large permutation space and conclude that input reversal is infeasible.

In this paper, we show that such permutation-based schemes are in general not secure in the open-weights setting. We devise a novel **vocab-matching attack**[1] (see Figure 1) that can nearly perfectly decode the user input in this setting, improving on existing work (Wan et al., 2024). We show that this attack maintains near-perfect decoding performance across various permutation types, including those used by the schemes above. Furthermore, the attack is capable of decoding against common noising methods proposed in the literature for private inference (Morris et al., 2023a).

We then introduce a new multi-party scheme, **Cascade** (see

---

[*]Equal contribution  [1]Ritual AI  [2]Stanford University  [3]Columbia University. Correspondence to: Arka Pal (Project Lead) <arka@ritual.net>.

*Proceedings of the 42nd International Conference on Machine Learning*, Vancouver, Canada. PMLR 267, 2025. Copyright 2025 by the author(s).

[1]Our implementation is available at https://github.com/ritual-net/vma-external.

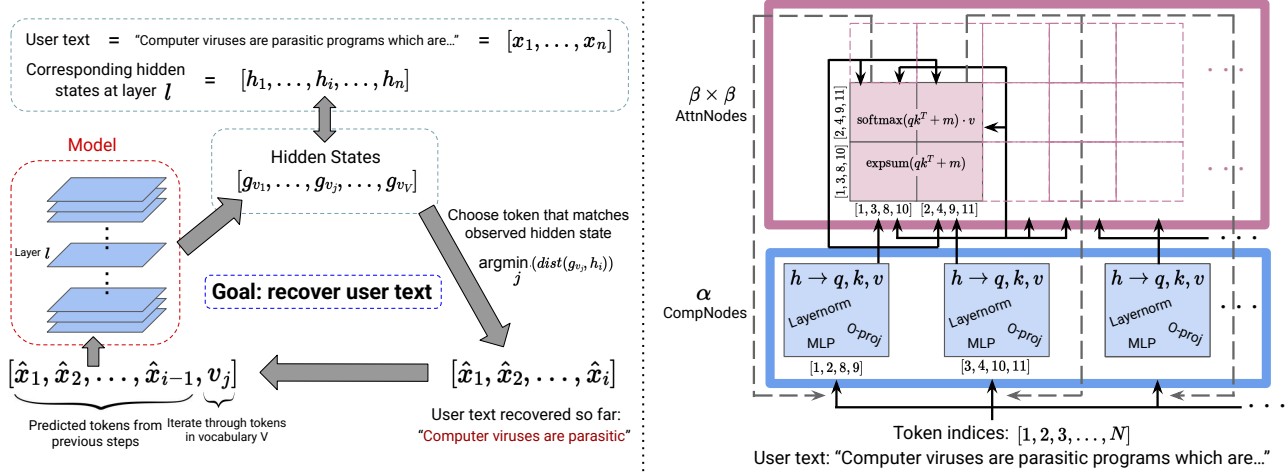

*Figure 1.* **Left:** High-level representation of the **vocab-matching attack** to decode user text from hidden states in the open-weights setting. This attack achieves nearly perfect decoding accuracy, even if hiddens are permuted or noised. **Right:** Schematic representation of our proposed privacy-preserving multi-party inference scheme, **Cascade**, which prevents the vocab-matching attack.

Figure 1), which avoids our attack by leveraging *token sharding*. We show that Cascade is also resistant to existing reversal approaches in the literature (Wan et al., 2024; Morris et al., 2023b). While Cascade does not provide the rigorous privacy guarantees of cryptographic MPC schemes, it is much more efficient than them, and presents a new paradigm in the trade-off between scalability and security.

## 2. Setup & Threat Model

We assume the setting of a user $U$ who wishes to perform inference with an LLM model $M$ on some input prompt $\boldsymbol{x}$, which can be considered as an ordered sequence of tokens $[x_1, x_2, ..., x_N]$. We denote the size of the hidden state of the LLM by $d$, and the sequence length by $N$.

As the user $U$ does not have the resources to perform the inference themselves, they rely on a set of third-parties $P_1, P_2, ..., P_K$. The model weights of $M$, including the embedding lookup table, are known to all parties. We assume that the parties behave *semi-honestly*, as in past works (Zheng et al., 2024; Luo et al., 2024; Dong et al., 2023b; Yuan et al., 2024). Semi-honest parties follow the defined protocol faithfully, but may attempt to recover the user's data from information that they receive during the protocol.

## 3. Related Work

Several existing works have investigated the reversibility of LLM embeddings (Song & Raghunathan, 2020; Morris et al., 2023a; Li et al., 2023b; Kugler et al., 2024) with relatively good decoding performance. Unlike our setting, these focus on the reversal of a single vector $\boldsymbol{e} = \phi(\boldsymbol{x}) \in \mathbb{R}^d$, where $\phi$ is an embedding model that returns a single fixed-size vector from an $N$-token input $\boldsymbol{x} = [x_1, x_2, ..., x_N]$. In

our paper, we are instead concerned with the reversibility of full intermediate LLM states $[h_1, h_2, ..., h_N] \in \mathbb{R}^{N \times d}$.

The two closest previous works on reversibility in our setting are those of Wan et al. (2024) and Morris et al. (2023b). The former deals with hidden state reversal, while the latter focuses on logit output distribution reversal. In both papers, the authors train a transformer-based network to reverse the sequence of hidden states into the original token inputs. Experiments are conducted on two decoder-based models, Llama-2-7B and ChatGLM-6B. Average F1 scores of approximately $60\%$ are achieved across a range of datasets in Wan et al. (2024) on hidden states near the last layers, and scores around $75\%$ are achieved for logit reversal in Morris et al. (2023b). Importantly, the latter paper does not assume adversary access to model weights, while the former deals with a model provider performing inference on users' embeddings and is more analogous to our setting.

Petrov et al. (2024) propose an attack that shares some elements with ours, like the exploitation of the unidirectional nature of decoder-based LLMs and the finite lookup table. However, they focus on gradient reversal into original inputs in the federated *training* setting, distinct from our private *inference* setting. Also, their method relies on full-rank properties of the gradients, which are not always satisfied. By contrast, our method does not have any such restrictions.

To the best of our knowledge, no existing work succeeds at reversing permutations of LLM hidden states.

## 4. Hidden State Reversal

We begin by considering the general case where one of the parties performing inference, $P_k$, receives hidden states $\boldsymbol{h} = [h_1, h_2, ..., h_N]$ at some layer $l$ of the LLM $M$.

Can the party $P_k$ reverse the hidden states $\boldsymbol{h}$ to the input sequence of tokens $\boldsymbol{x} = [x_1, x_2, ..., x_N]$ that produced $\boldsymbol{h}$?

### 4.1. Informal Attack Description

We outline our proposed attack below, and provide a visual depiction in Figure 1.

The attack begins with a batched forward pass over all length-1 sequences $[v]$, where $v$ ranges over words in the vocabulary $\mathcal{V}$. From this, the adversary gets $V = |\mathcal{V}|$ candidate layer $l$ hidden states $\boldsymbol{h}(v) \in \mathbb{R}^{1 \times d}$. They set the first predicted input token $\widehat{x}_1$ to be the token $v$ for which $\boldsymbol{h}(v)$ exactly matches the first hidden state $h_1$.

Next, the adversary performs a batched forward pass over all length-2 sequences $[\widehat{x}_1, v]$ with $v \in \mathcal{V}$, to get $V$ candidate layer $l$ hidden states $\boldsymbol{h}(\widehat{x}_1, v) \in \mathbb{R}^{2 \times d}$. Now, they set the second predicted input token $\widehat{x}_2$ to be the token $v$ where the second row of $\boldsymbol{h}(\widehat{x}_1, v)$ equals the second hidden state $h_2$.

In general, at the $n$th stage, using the first $n-1$ predicted input tokens $\widehat{x}_1, \ldots, \widehat{x}_{n-1}$, the adversary performs a forward pass over all length-$n$ sequences $[\widehat{x}_1, \ldots, \widehat{x}_{n-1}, v]$ with $v \in \mathcal{V}$. They obtain $V$ candidate layer $l$ hidden states $\boldsymbol{h}(\widehat{x}_1, \ldots, \widehat{x}_{n-1}, v) \in \mathbb{R}^{n \times d}$, and set the $n$th predicted input token $\widehat{x}_n$ to be the token $v$ where the $n$th (last) row of candidate states matches the $n$th hidden state $h_n$. Iterating over $n = 1, \ldots, N$, the adversary sequentially obtains the predicted input sequence $\widehat{\boldsymbol{x}}$ from the layer $l$ hidden states $\boldsymbol{h}$.

Naively, one might expect that an exact match of $\boldsymbol{h}$ would require an exponential brute-force search over all $V^N$ possible sequences of tokens $\boldsymbol{x}$. However, we see that by exploiting the autoregressive property of transformers, this is reduced to a linear search; the total cost of this attack is $O(VN)$.

### 4.2. Practical Implementation

#### 4.2.1. Assumptions

The key assumptions necessary for our attack are as follows.

**(A1)** The forward pass over the vocabulary in the attack will match the forward pass that generated the given hiddens $\boldsymbol{h}$.

**(A2)** Hidden states of LLMs are *non-colliding*: there are few matches between candidate tokens $v$ and hidden states $h_n$ at each step of the attack. If the average number of matches at each step is $M$, then the search space grows as $M^N$, which is infeasible when $M$ or $N$ is large. Prior work (Dong et al., 2023a) has demonstrated the rank-reducing effects of attention, so it is plausible that the size of the subspace in latter layers is too small to prevent collisions. Still, we will see this assumption holds in practice.

**(A3)** The LLM employs *unidirectional* attention. This holds for decoder-only architectures, which are de-facto for many current state-of-the-art LLMs.

**(A4)** The model weights are known to the party $P_k$. Later, in the settings of Yuan et al. (2024); Luo et al. (2024), we can relax this assumption.

Generally, **(A1)** is not satisfied due to **non-determinism**.

#### 4.2.2. Non-Determinism

Assumption **(A1)** asserts the forward pass performed over the vocabulary matches the forward pass that generated the given hiddens exactly. In general, due to the non-associativity of floating-point operations (Villa et al., 2009), this does not hold. In the GPU setting with parallel asynchronous thread execution and pooling without global synchronization, there can be considerable variation in the output (Shanmugavelu et al., 2024). In addition, differences in hardware, random number seeds, environment variables, and the state of initialized memory on the machine add to the variability, and these may not be known to the adversary.

Due to the presence of this reducible and irreducible noise, exact matching cannot be used successfully with our attack. Thus, we loosen our matching requirements by computing the *L1-distance* between the last row of candidate hidden states and the given hidden state, and accept a match for a token if the distance is below some threshold $\epsilon$. If no match is found, we choose the token with minimal L1-distance.

However, by allowing an $\epsilon$-ball for matching, we increase the possibility of collisions as stated above. Is our attack still successful – i.e. is assumption **(A2)** still satisfied – even with this fuzzy matching? In Section 4.4, we find the answer is emphatically yes.

#### 4.2.3. Efficiency

We optimize runtime using a *proposal model* to provide a likelihood-based order of iteration through the vocabulary. We find that this modification reduces the average number of tokens searched through at each step from $V/2$ to $\sim 100$, resulting in over a $1000\times$ speedup. Further, we implement a novel variation of key-value-caching (KV-caching) to reduce the computational cost of our attack. Further details on these optimizations are given in Appendix A. With these improvements, we reduce the decoding time of prompts of length 50 from many hours to typically less than 30 seconds. We provide details on the scalability of our attack with respect to model size in Appendix B.

### 4.3. Formal Attack Description

We now provide a formalized description of our attack, incorporating the modifications for efficiency and handling the nondeterminism described above, in Algorithm 1. For a step-by-step walkthrough, see Appendix C.

**Algorithm 1** Vocabulary-Matching Attack

---

**input** Model $M$, layer $l$ hidden states $\boldsymbol{h} = [h_1, \ldots, h_N]$, vocabulary $\mathcal{V}$, proposal model $P$, L1-threshold $\epsilon$
**output** Decoded token sequence $\widehat{\boldsymbol{x}} = [\widehat{x}_1, \widehat{x}_2, \ldots, \widehat{x}_N]$

1: Initialize empty sequence $\widehat{\boldsymbol{x}} \leftarrow []$
2: **for** $i = 1$ to $N$ **do**
3:    $\mathcal{V}_{\text{ordered}} \leftarrow \text{argsort}(P(\widehat{\boldsymbol{x}}))$ {Get ordered vocabulary from proposal model}
4:    min_dist $\leftarrow \infty$
5:    best_match $\leftarrow$ None
6:    **for** $v \in \mathcal{V}_{\text{ordered}}$ **do**
7:       $g \leftarrow M_{\leq l}([\widehat{\boldsymbol{x}}, v])$ {Forward pass up to layer $l$}
8:       dist $\leftarrow \|g - h_i\|_1$ {Calculate L1 distance}
9:       **if** dist $<$ min_dist **then**
10:          min_dist $\leftarrow$ dist
11:          best_match $\leftarrow v$
12:       **end if**
13:       **if** dist $< \epsilon$ **then**
14:          $\widehat{x}_i \leftarrow v$
15:          break
16:       **end if**
17:    **end for**
18:    **if** dist $\geq \epsilon$ **then**
19:       $\widehat{x}_i \leftarrow$ best_match
20:    **end if**
21: **end for**
22: **return** $\widehat{\boldsymbol{x}}$

---

### 4.4. Experiments & Discussion

We conduct our experiments on two state-of-the-art open-source LLMs, Gemma-2-2B-IT (Team et al., 2024) and Llama-3.1-8B-Instruct (Grattafiori et al., 2024). These models have different sizes, training methods, and architectures. We test on samples from the Fineweb-Edu dataset (Penedo et al., 2024). The proposal model is the same as the model being attacked. To ensure that the dataset is unseen by the proposal model, we use the CC-MAIN-2024-10 data split, which postdates the models' training cutoff dates. To speed up evaluation, we test on every fifth layer of the model, extracting hidden states at layers 1, 6, 11, 16, 21, and 26.

For each layer of interest, we tune $\epsilon$ by performing a ternary search on a small training set of 50 prompts from FineWeb, to determine the optimal L1-threshold under which predicted tokens are accepted as matches. We evaluate on 1000 held out prompts, and our results are shown in Table 1. We find that nearly all evaluation samples are perfectly decoded. All $\epsilon$ values are given in Appendix D. Due to computational constraints, each evaluation prompt was truncated to a maximum of 50 tokens; however, small-scale experiments with prompts over 200 tokens demonstrated that our results generalize to longer prompt settings – vocab-matching still perfectly decodes hidden states into their input tokens.

*Table 1.* Percentage of perfectly decoded evaluation samples by vocab-matching at layers of Gemma-2-2B-IT and Llama-3.1-8B-Instruct.

| Layer | Gemma | Llama |
|-------|-------|-------|
| 1     | 100%  | 100%  |
| 6     | 100%  | 100%  |
| 11    | 100%  | 100%  |
| 16    | 100%  | 100%  |
| 21    | 100%  | 99.9% |
| 26    | 100%  | 99.7% |

In the rare cases where perfect decoding was not achieved, prompts contained formatting characters like newlines and hyphens, or grammatically incorrect word repetitions. These cause the proposal model to accept an incorrect token early when its L1 error is below the $\epsilon$ threshold, even when the correct token has a small L1 error. Tuning $\epsilon$ with a larger training set or using a full vocabulary search may mitigate these errors, but at a higher computational cost.

The success of our attack allows us to conclude that LLM hidden states are highly distinct and non-colliding.

## 5. Permuted Hidden State Reversal

We now consider the case where one of the parties performing inference receives a permutation of the intermediate sequence of hidden states $\boldsymbol{h}$ at some layer $l$ of the LLM $M$.

### 5.1. Existing Work

Recently, a number of works have proposed utilizing permutations to perform privacy-preserving inference in a multi-party-computation (MPC) setup.

Zheng et al. (2024) introduce **PermLLM**. They permute at the non-linear components of the LLMs in order to reveal them 'safely' to one of the parties, and therefore avoid expensive iterated inter-party communication. The permutation is done on the attention logits before the softmax, at layer normalizations, and at the non-linear functions in the MLP block. The latter is a purely elementwise function, so the authors can do a full permutation across the $[N, d]$ elements, resulting in a permutation space of size $(Nd)!$. However, softmax and layer-norm are row-wise operations, so the permutation applied in this case is a (distinct) permutation to the columns, followed by a permutation of the $N$ rows, resulting in a permutation space of size $N!(d!)^N$.

Yuan et al. (2024) introduce **STIP**. They permute both the model weights and the user prompt embeddings in the hidden $d$ dimension, and the entire inference process (on the next token) is then carried out by a single party.

Luo et al. (2024) apply ideas from both of the above works to formulate **Centaur**. This permutes the model weights, utilizing additive secret-sharing for the linear layers, but relies on two-party permuted plaintext computation at the non-linearities (softmax, layer-norm and GeLU). Permutation is applied in the hidden $d$ dimension.

## 5.2. Permuted Intermediate States Are Not Safe

We now propose a modification of our vocab-matching attack in Section 4, which breaks user input privacy for the above schemes in the open-weight setting. Extensions to the attack also break privacy in the closed-weights setting that Yuan et al. (2024) and Luo et al. (2024) originally consider: see Appendix I for further details. Also, in Appendix J, we explain why theoretical claims on statistical permutation security from Luo et al. (2024) do not anticipate our attack.

We present our modified attack in the sequence-dimension, hidden-dimension, and factorized-2D permutation settings. See Appendix E for detailed pseudocode of the modifications. Our experimental setup is the same as in Section 4.4.

### 5.2.1. Sequence-Dim Permutation

Assume that permutation has been applied to layer $l$ hidden states $\boldsymbol{h} = [h_1, h_2, ..., h_N]$ in the sequence dimension:

$$\boldsymbol{h}_{\text{seq\_perm}} = [h_{\sigma(1)}, h_{\sigma(2)}, ..., h_{\sigma(N)}]$$

where $\sigma$ is a permutation of $[N] = \{1, 2, \ldots, N\}$. Then, we modify the vocab-matching attack as follows. At the $n$th stage, we now choose the vocabulary token $v$ where the $n$th row of the corresponding candidate hidden state is within an L1-distance of $\epsilon$ from *any* row of $\boldsymbol{h}_{\text{seq\_perm}}$. Suppose this $\epsilon$-ball match is made with the $i$th row $h_{\sigma(i)}$. We set the $n$th predicted input token $\widehat{x}_n$ to $v$, and remove $h_{\sigma(i)}$ from consideration for hidden state matching in *all* future stages. Iterating over $n = 1, \ldots, N$, we obtain the predicted input sequence $\widehat{\boldsymbol{x}}$ from sequence-permuted hidden states $\boldsymbol{h}_{\text{seq\_perm}}$.

Compared to the vocab-matching attack, the opportunities for collision are now increased up to $N$-fold, as we match with up to $N$ rows of $\boldsymbol{h}$ rather than one. However, we again observe very few collisions in practice and are able to decode essentially all input prompts: see Table 2.

### 5.2.2. Hidden-Dim Permutation

Next we consider the case where permutation has been performed on the hidden dimension of $\boldsymbol{h}$ instead. That is, the party performing inference is now given:

$$\boldsymbol{h}_{\text{hidden\_perm}} = [\pi_1(h_1), \pi_2(h_2), ..., \pi_N(h_N)]$$

where each $\pi_i$ permutes elements of a $d$-dimensional vector. In this setting, it is no longer possible to use L1-distance directly to find the nearest vocabulary token match. We

instead use the **sorted L1-distance**, which individually sorts the two vectors to be compared and then computes their L1-distance. Again, the existence of noise may appear to be a significant obstacle to this approach. However, we find that even this relatively simple matching approach is robust enough to noise to achieve nearly perfect decoding. Our results are shown in Table 2.

### 5.2.3. Factorized-2D Permutation

We now consider the case of a factorized two-dimensional permutation as used in Zheng et al. (2024), where a hidden-dimension permutation is applied to each hidden state, and then these resulting states are shuffled in the sequence dimension. The adversary now has:

$$\boldsymbol{h}_{\text{fact\_perm}} = [\pi_1(h_{\sigma(1)}), \pi_2(h_{\sigma(2)}), ..., \pi_N(h_{\sigma(N)})]$$

where $\sigma$ is a permutation of $[N]$ and each $\pi_i$ permutes a $d$-dimensional vector. The attack in this setting again utilizes the sorted-L1 matching function, but now expands to consider all $N$ rows of $\boldsymbol{h}_{\text{fact\_perm}}$, as in Section 5.2.1. Remarkably, even in this setting, the hidden states of both models are decoded nearly perfectly across layers (Table 2).

We conclude that permuted hidden states of LLMs are highly decodeable by our attack, and therefore schemes which expose them are not secure in the open-weights setting.

*Table 2.* The percentage of evaluation samples that were perfectly decoded under sequence-dim, hidden-dim, and factorized-2D permutations, for Gemma-2-2B-IT and Llama-3.1-8B-Instruct.

| Layer | Sequence-Dim | | Hidden-Dim | | Factorized-2D | |
|---|---|---|---|---|---|---|
| | Gemma | Llama | Gemma | Llama | Gemma | Llama |
| 1 | 100% | 99.7% | 100% | 100% | 99.9% | 98.4% |
| 6 | 99.8% | 100% | 100% | 98.5% | 99.5% | 97.8% |
| 11 | 100% | 100% | 100% | 99.2% | 99.5% | 98.9% |
| 16 | 100% | 100% | 99.9% | 99.4% | 99.2% | 98.8% |
| 21 | 99.8% | 100% | 98.2% | 98.9% | 99.1% | 98.0% |
| 26 | 99.8% | 100% | 98.0% | 98.2% | 99.0% | 97.6% |

## 6. Noised & Quantized Hidden State Reversal

Section 5 shows that modifications to our attack can decode sequence-dimension, hidden-dimension, and factorized-2D permutations of hidden states. We now examine the efficacy of our attack on methods of defense which aim to disrupt assumption **(A2)** in Section 4.2.1, by noising the hidden states or quantizing the model. With enough noise or quantization, decoding can be made impossible. However, high noise will likely disrupt LLM performance. Thus, any such defense is a delicate balancing act of ensuring security against our attack, while still maintaining downstream model performance. We find that generally, these methods are still not sufficient to defend against our attack. Due to space constraints, we provide much further detail in Appendix F.

# 7. Token-Sharded Multi-Party Inference

As we have found that permutations of hidden states are not secure, a natural follow-up question is whether *sharded* hidden states are secure. We suggest the answer is affirmative for certain sharding schemes. We propose a defense to the vocab-matching attack based on *token-dimension sharding* of hidden states, which leads to a new multi-party inference scheme: **Cascade**.

Notably, Cascade does not use cryptographic primitives; the computations are nearly unchanged from a standard forward pass, so almost no additional computational overhead is incurred. There is also no degradation of performance, as unlike SMPC schemes, no approximations need to be made to efficiently compute non-linearities. The scheme also does not require any trusted party interaction during inference.

## 7.1. Scheme Description

At a high level, Cascade exploits the fact that only the self-attention mechanism in transformers has interaction among tokens in a sequence; for all other parts of the architecture, the tokens are treated like batch dimension elements.

Therefore, the *CompNodes* each receive only a subset of the tokens, and perform the bulk of batch operations in the LLM. Then, self-attention is performed by separate *AttnNodes*, who each receive $Q, K, V$-projections of sharded hidden states from the CompNodes. The AttnNodes send partial attention outputs at each layer back to the CompNodes, who can reconstruct shards of the full attention output.

**Sharding** In multi-headed attention, we let $H$ and $H_{KV}$ be the attention head and key-value head (for grouped-query attention) counts, $N$ be the token count, $d_{emb}$ be the hidden dimension, and $d$ be the attention hidden dimension. There are three axes of sharding used along the token dimension: **(1)** the sharding of hidden states $\boldsymbol{h} \in \mathbb{R}^{N \times d_{emb}}$, **(2)** the sharding of query states $\boldsymbol{q} \in \mathbb{R}^{H \times N \times d}$, and **(3)** the sharding of key and value states $\boldsymbol{k}, \boldsymbol{v} \in \mathbb{R}^{H_{KV} \times N \times d}$. These involve splitting token indices $[N] = \{1, 2, \dots, N\}$ into a union of disjoint subsets $\{R_i\}_{i=1}^{\alpha}$ for hiddens, $\{S_j\}_{j=1}^{\beta}$ for queries, and $\{T_k\}_{k=1}^{\gamma}$ for keys and values, where $\alpha, \beta, \gamma$ are shard counts. We also shard the positional embeddings $\boldsymbol{p} \in \mathbb{R}^{N \times d_{emb}}$ and the attention mask $\boldsymbol{s} \in \mathbb{R}^{H \times N \times N}$, which are initialized pre-inference, as well as the masked logits $\boldsymbol{a} \in \mathbb{R}^{H \times N \times N}$ and attention output $\boldsymbol{o} \in \mathbb{R}^{N \times d_{emb}}$.

**Symmetrization** In Appendix M, we show that we can make $S$ and $T$ sharding equal at no loss of security. Thus, from here on, we replace $T$ with $S$ and $\gamma$ with $\beta$.

Our symmetrized notation for any layer is in Table 3. Note that the last six rows rely on tensors that are not directly shards of LLM states, but are derived from them, as in Table 4. Here, max, softmax, and expsum are performed row-wise, with $\operatorname{expsum}(\boldsymbol{x}) := \sum_i \exp(x_i)$. Also, $\boldsymbol{e}^{S_k}$ only performs an expsum after subtracting row-wise maximums, as in the numerically stable subtract-max form of softmax.

| Tensor | Shape | Shards | Notation |
|---|---|---|---|
| $\boldsymbol{h}, \boldsymbol{p}, \boldsymbol{o}$ | $(N, d_{emb})$ | $(R_i, *)$ | $\boldsymbol{h}^{R_i}, \boldsymbol{p}^{R_i}, \boldsymbol{o}^{R_i}$ |
| $\boldsymbol{q}, \boldsymbol{k}, \boldsymbol{v}$ | $(H, N, d)$ | $(*, R_i, *)$ | $\boldsymbol{q}^{R_i}, \boldsymbol{k}^{R_i}, \boldsymbol{v}^{R_i}$ |
| $\boldsymbol{q}, \boldsymbol{k}, \boldsymbol{v}$ | $(H, N, d)$ | $(*, S_j, *)$ | $\boldsymbol{q}^{S_j}, \boldsymbol{k}^{S_j}, \boldsymbol{v}^{S_j}$ |
| $\boldsymbol{q}, \boldsymbol{k}, \boldsymbol{v}$ | $(H, N, d)$ | $(*, R_i \cap S_j, *)$ | $\boldsymbol{q}^{R_i \cap S_j}, \boldsymbol{k}^{R_i \cap S_j}, \boldsymbol{v}^{R_i \cap S_j}$ |
| $\boldsymbol{a}, \boldsymbol{s}$ | $(H, N, N)$ | $(*, *, S_k)$ | $\boldsymbol{a}^{S_k}, \boldsymbol{s}^{S_k}$ |
| $\boldsymbol{a}, \boldsymbol{s}$ | $(H, N, N)$ | $(*, R_i, S_k)$ | $\boldsymbol{a}^{R_i S_k}, \boldsymbol{s}^{R_i S_k}$ |
| $\boldsymbol{a}, \boldsymbol{s}$ | $(H, N, N)$ | $(*, S_j, S_k)$ | $\boldsymbol{a}^{S_j S_k}, \boldsymbol{s}^{S_j S_k}$ |
| $\boldsymbol{a}, \boldsymbol{s}$ | $(H, N, N)$ | $(*, R_i \cap S_j, S_k)$ | $\boldsymbol{a}^{(R_i \cap S_j) S_k}, \boldsymbol{s}^{(R_i \cap S_j) S_k}$ |
| $\boldsymbol{m}^{S_k}, \boldsymbol{e}^{S_k}$ | $(H, N, 1)$ | $(*, R_i, *)$ | $\boldsymbol{m}^{R_i S_k}, \boldsymbol{e}^{R_i S_k}$ |
| $\boldsymbol{m}^{S_k}, \boldsymbol{e}^{S_k}$ | $(H, N, 1)$ | $(*, S_j, *)$ | $\boldsymbol{m}^{S_j S_k}, \boldsymbol{e}^{S_j S_k}$ |
| $\boldsymbol{m}^{S_k}, \boldsymbol{e}^{S_k}$ | $(H, N, 1)$ | $(*, R_i \cap S_j, *)$ | $\boldsymbol{m}^{(R_i \cap S_j) S_k}, \boldsymbol{e}^{(R_i \cap S_j) S_k}$ |
| $\boldsymbol{u}^{S_k}$ | $(H, N, |S_k|)$ | $(*, R_i, *)$ | $\boldsymbol{u}^{R_i S_k}$ |
| $\boldsymbol{u}^{S_k}$ | $(H, N, |S_k|)$ | $(*, S_j, *)$ | $\boldsymbol{u}^{S_j S_k}$ |
| $\boldsymbol{u}^{S_k}$ | $(H, N, |S_k|)$ | $(*, R_i \cap S_j, *)$ | $\boldsymbol{u}^{(R_i \cap S_j) S_k}$ |

*Table 3.* Sharding for intermediate LLM states, where $*$ is a full slice. In slicing notation, $\boldsymbol{a}^{(R_i \cap S_j) S_k} = \boldsymbol{a}[:, R_i \cap S_j, S_k]$ and $\boldsymbol{m}^{(R_i \cap S_j) S_k} = \boldsymbol{m}^{S_k}[:, R_i \cap S_j, :]$, with $\boldsymbol{m}^{S_k}$ as in Table 4.

| Tensor | Shape | Formula |
|---|---|---|
| $\boldsymbol{m}^{S_k}$ | $(H, N, 1)$ | $\max(\boldsymbol{a}^{S_k})$ |
| $\boldsymbol{e}^{S_k}$ | $(H, N, 1)$ | $\operatorname{expsum}(\boldsymbol{a}^{S_k} - \boldsymbol{m}^{S_k})$ |
| $\boldsymbol{u}^{S_k}$ | $(H, N, |S_k|)$ | $\operatorname{softmax}(\boldsymbol{a}^{S_k}) \boldsymbol{v}^{S_k}$ |

*Table 4.* Tensors derived from sharded states $\boldsymbol{a}^{S_k}, \boldsymbol{v}^{S_k}$ in Table 3.

**Nodes** There are two types of nodes which hold the sharded states above: **CompNodes** and **AttnNodes**. We initialize $\alpha$ CompNodes and $\beta^2$ AttnNodes, indexed as CompNode$_i$ and AttnNode$_{jk}$ for all $i \in [\alpha]$ and $j, k \in [\beta]$.

**Inference** Cascade breaks single layer inference into the **pre-pass** by CompNodes, **attention-pass** by AttnNodes, and **post-pass** by CompNodes. We describe these in Algorithm 7, Algorithm 8, and Algorithm 9 in Appendix G. These form a single layer pass in Algorithm 2.

During Cascade layer $l$ inference, each CompNode$_i$ begins with $\boldsymbol{h}^{R_i}$ at layer $l$, and ends with $\boldsymbol{h}^{R_i}$ at layer $l + 1$. This output is the input to CompNode$_i$ for layer $l + 1$ inference, so inference for all layers can follow Algorithm 2.

After the last layer, CompNodes apply the LM head to get $R_i$-sharded logits, and the CompNode with the last token will use this to generate the next token's embedding[2]. This is sent back to the CompNode with the next token index in $R_i$. Then, Cascade inference repeats to generate the next token. This can be sped up with KV-caching: see Appendix Q.

---

[2]The logits can also be returned to the user if desired.

**Algorithm 2** Cascade Single Layer Forward Pass

---

**input** $h^{R_i}$ at layer $l$ from CompNode$_i$ for $1 \leq i \leq \alpha$
**output** $h^{R_i}$ at layer $l+1$ to CompNode$_i$ for $1 \leq i \leq \alpha$

1: **for** $i = 1$ to $\alpha$: CompNode$_i$ **do**
2:     $q^{R_i}, k^{R_i}, v^{R_i} \leftarrow$ **pre_pass**$(h^{R_i})$
3:     **for** $j, k = 1$ to $\beta$: AttnNode$_{jk}$ **gets**
4:         $q^{R_i S_j}, k^{R_i S_k}, v^{R_i S_k}$
5:     **end for**
6: **end for**
7: **for** $j, k = 1$ to $\beta$: AttnNode$_{jk}$ **do**
8:     $m^{S_j S_k}, e^{S_j S_k}, u^{S_j S_k} \leftarrow$ **attn_pass**$(q^{S_j}, k^{S_k}, v^{S_k})$
9:     **for** $i = 1$ to $\alpha$: CompNode$_i$ **gets**
10:        $m^{(R_i \cap S_j)S_k}, e^{(R_i \cap S_j)S_k}, u^{(R_i \cap S_j)S_k}$
11:     **end for**
12: **end for**
13: **for** $i = 1$ to $\alpha$: CompNode$_i$ **do**
14:     $o^{R_i} \leftarrow$ **post_pass**$\{m^{R_i S_k}, e^{R_i S_k}, u^{R_i S_k}\}_{k=1}^{\beta}$
15:     $h^{R_i} \leftarrow$ **mlp**$(h^{R_i} + o^{R_i})$ {Residual and MLP}
16: **end for**

---

## 7.2. Security Analysis

In this section, we examine the security properties of Cascade. Cascade does not employ cryptographic techniques, so we can only elucidate on statistical security. Nevertheless, we examine a wide range of security considerations below. The security of Cascade is a function of its implementation parameters: the *number of nodes* participating, as well as the *sharding strategy* used, i.e. $\{R_i\}_{i=1}^{\alpha}$ and $\{S_j\}_{j=1}^{\beta}$.

**Information Leakage** To analyze information leakage, we examine isolated computational stages of nodes. In all stages (pre-pass, attention-pass, or post-pass) of Cascade, no information is received *during* the computation. Therefore, all input leakage comes from the stage initialization. Thus, any leakage must occur from the following sharded tensors:

$$\text{CompNode}_i \rightarrow h^{R_i}, \{m^{R_i S_k}, e^{R_i S_k}, u^{R_i S_k}\}_{k=1}^{\beta},$$
$$\text{AttnNode}_{jk} \rightarrow q^{S_j}, k^{S_k}, v^{S_k}.$$

We will examine leakage to CompNodes from $h^{R_i}$ in Section 7.2.1 and Section 7.2.2. We will analyze leakage from $q^{S_j}, k^{S_k}, v^{S_k}$ in Section 7.2.3. Finally, we consider all other sources of leakage to CompNodes in Section 7.2.4.

**Sharding Strategies** When more nodes participate, each has access to fewer tokens and hidden states, which makes reversing the full input more difficult. Even holding the node count fixed, the choice of sharding may also affect the difficulty of reversal. For example, is it more secure for a node to have access to $[h_1, h_5, h_9, h_{13}]$ or $[h_1, h_2, h_{12}, h_{13}]$?

As the space of sharding strategies is vast, we focus on $(c, \delta)$**-sharding strategies**, where each sharded index set takes the form of a 'clustered-arithmetic' or $(c, \delta)$-sequence.

**Definition 7.1.** We say a subset of indices $[N]$ is a $(c, \delta)$-sequence if it takes the following form for some $i$:

$$\{i, i+1, \ldots, i+c-1, \delta+i, \delta+i+1, \ldots, \delta+i+c-1,$$
$$2\delta+i, 2\delta+i+1, \ldots, 2\delta+i+c-1, \ldots\}.$$

We focus on these strategies as they fulfill several security desiderata. We emphasize however that other strategies may be preferable to $(c, \delta)$-sharding depending on the use-case.

$(c, \delta)$ **vs. Number of CompNodes** Under $(c, \delta)$-sharding, the minimum number of CompNodes $\alpha$ needed to ensure all indices in $[N]$ are held by some node is given by $\alpha = \lceil \delta/c \rceil$.

### 7.2.1. LAYER 0 COMPNODE HIDDEN SECURITY

We first consider the security of CompNodes at layer 0 of the scheme, who each hold some token embeddings of the input prompt $x$. As embeddings are immediately reversible to their tokens through the lookup table, we conclude that Cascade should not be used when the security of every token is paramount. If individual token security is imperative, Cascade can be integrated with SMPC as in Appendix H.

Given that a CompNode has access to some of the $N$ token embeddings, say $[e_{n_1}, e_{n_2}, \ldots, e_{n_t}]$, the possibility of reconstruction of the full prompt $x$ is theoretically lower bounded by the entropy of the distribution

$$p(\{x_j : j \in [N] \setminus \{n_1, \ldots, n_t\}\} \mid x_{n_1}, x_{n_2}, \ldots, x_{n_t})$$

The true distribution cannot easily be computed, but we may approximate it using masked token infilling, as in the training of models like BERT (Devlin et al., 2019), RoBERTa (Liu et al., 2019), and ModernBERT (Warner et al., 2024). We use the recently released state-of-the-art ModernBERT-large to probe properties of this distribution under $(c, \delta)$-sharding. We use 200 samples from FineWeb-Edu to compute the ROUGE-L (Lin, 2004) score over argmax-token generation. Our results are shown in Figure 2. We see that ROUGE-L score diminishes as both $c$ and $\alpha$ increase; good security seems to be achieved for $c, \alpha \geq 8$ and thus $\delta \geq 64$.

### 7.2.2. LAYER > 0 COMPNODE HIDDEN SECURITY

We now turn our attention to CompNode security of $h^{R_i}$ at deeper layers of the LLM. Can we select a sharding strategy that defends against our attack outlined in Section 4? Also, do we remain secure to learning-based attacks as in prior works Wan et al. (2024) and Morris et al. (2023b)?

**Definition 7.2.** Let $t_{\max}$ be the maximum value of $t$ for which an adversary has resources to perform $V^t$ forward passes. Then we define the **vocab-matching threshold** $\rho$ as $\rho := t_{\max} + 1$.

**Vocab-Matching Attack** Our defense against vocab-matching is to ensure large token gaps in nodes. For a

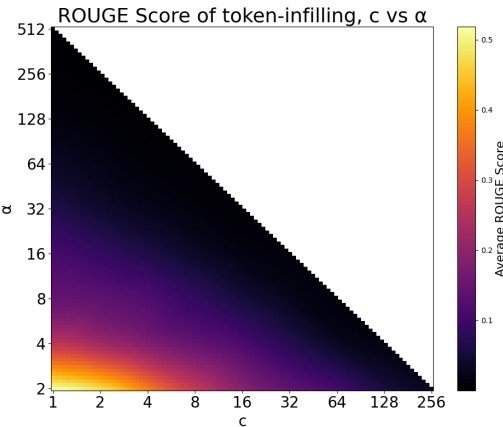

*Figure 2.* ROUGE-L scores for Layer 0 token prediction using ModernBERT-Large, as a function of $c$, the number of 'clusters' in the sharding scheme, and $\alpha$, the number of CompNodes. Higher $\alpha$ and higher $c$ tend towards lower ROUGE-L, increasing security.

$(c, \delta)$-sharding scheme, in each shard, the distance from one 'cluster' of indices to the next is $\delta - c + 1$. Thus, in order to carry out the attack, an attacker cannot perform a single run through $\mathcal{V}$ and obtain the next token match; they must search over length $\delta - c + 1$ sequences of infilled words, like in Appendix C. The cost scales exponentially as $V^{\delta-c+1}$. Because $V \sim O(100000)$ in typical modern LLMs[3] the vocab-matching threshold of any adversary is likely no more than $\rho = 3$ in practice. Note $\delta - c + 1 = (\alpha - 1)c + 1$, so regardless of the value of $\rho$, we can increase $\alpha$ or $c$ until $\delta - c + 1 \geq \rho$ to prevent vocab-matching. For $\rho = 3$, all $\alpha, c \geq 2$ satisfy this.

**Learning-Based Attacks**  We now consider learning-based reversal attacks. Although hidden states may not be as easily reversible as token embeddings, it may be the case that as attention propagates information among hidden states, the text-infilling task is easier at deeper layers of the LLM than at the purely textual level. We examine this hypothesis by fine-tuning Gemma-2-2B-IT and Llama-3.1-8B-Instruct on $(c, \delta)$-masked input sequences from FineWeb-Edu, with the target labels being the full input sequence. Note this is a 'worst-case' scenario where the adversary knows the $c$ and $\delta$ parameters. We update both models to use a bidirectional mask, in line with the token-infilling nature of this task. We train until the evaluation loss on a held-out set converges over layer 1 representations from each model, and evaluate on layer 1 hidden states. Our approach is similar to Wan et al. (2024); Morris et al. (2023b).

Our ROUGE-L scores for reconstructed Gemma-2-2B-IT prompts are shown in Table 5. In Appendix K, we simi-

---

[3]Although knowledge of the input distribution can make the base of the exponential smaller than $V$, the cost still scales exponentially, so large enough $\delta - c + 1$ prevents the attack variant.

larly analyze BLEU reconstruction scores (Papineni et al., 2002). When $c, \alpha \geq 8$, we have a ROUGE-L score less than 0.25, indicating significant reconstruction difficulty. We also tested on $c = 4, \alpha = 16$ and $c = 8, \alpha = 24$ and obtained ROUGE-L scores of 0.173 and 0.144, supporting that security continues to improve by scaling $c$ and $\alpha$.

*Table 5.* ROUGE-L scores of text reconstruction from the hiddens of layer 1 of Gemma-2-2B-IT for various values of $c$ and $\alpha$ under $(c, \delta)$-sharding. Increasing $c$ or $\alpha$ results in worse reconstruction.

|       | $\alpha = 4$ | $\alpha = 8$ | $\alpha = 12$ |
|-------|--------------|--------------|---------------|
| $c = 1$ | 0.701 | 0.467 | 0.349 |
| $c = 4$ | 0.427 | 0.290 | 0.230 |
| $c = 8$ | 0.355 | 0.222 | 0.191 |

We examine the choice of hidden layer, and if Llama representations are decoded better, in Appendix L. We find similar or better security across these other parameter choices.

### 7.2.3. ATTNNODE FULL SECURITY

So far, we analyzed CompNode security. In Appendix M, we examine AttnNode security and show how to improve it through $m$-splitting, which makes $S$ sharding more granular.

### 7.2.4. COMPNODE FULL SECURITY

In Section 7.2.1 and Section 7.2.2, we considered leakage to CompNodes from $\boldsymbol{h}^{R_i}$. Now, we consider how the shards $\boldsymbol{m}^{R_i S_k}, \boldsymbol{e}^{R_i S_k}, \boldsymbol{u}^{R_i S_k}$ impact security. Our main result is a condition *necessary* to prevent a modified vocab-matching attack. For a proof of this theorem, see Appendix P.

**Theorem 7.3.** *Suppose $\boldsymbol{s}$ is unidirectional. For Cascade to be secure, each gap between clusters of consecutive indices in each $R_i$ must have size $\geq \rho$.*

Note $(c, \delta)$-sharding satisfies this for $\delta \geq c + \rho$. In Appendix P, we argue further that such $(c, \delta)$-sharding is *sufficient* for *Layer 0* security, by demonstrating a reduction to an intractable linear program.

### 7.3. Communication and Computational Costs

We now enumerate communication and computational costs associated with Cascade. As in previous SMPC works (Li et al., 2023a; Dong et al., 2023b; Li et al., 2024), we assume perfect parallel transport in communication, a homogeneous node-wise bandwidth of $B$, and an inter-node latency of $\tau$. We denote $F$ as the number of bytes per element.

### 7.3.1. THEORETICAL COSTS

We first provide theoretical estimates of costs in Cascade.

**Computation**  There is almost no floating point overhead

in Cascade relative to vanilla inference: see Appendix N. We provide empirical evidence for this in Section 7.3.2.

**Communication**    We present single layer communication byte and time overhead formulae, derived in Appendix O:

$$\text{CommBytes} = \beta F(2dH + 2dH_{KV} + 2H) \cdot N \quad (1)$$

$$\begin{aligned}\text{CommTime} = 2\tau &+ \frac{\beta F d(H + 2H_{KV})}{B} \cdot \max_i |R_i| \\ &+ \frac{F(d+2)H}{B} \cdot \max_j |S_j|.\end{aligned} \quad (2)$$

Note Equation (1) scales linearly with $\beta$, so byte overhead is minimized with fewest AttnNodes. Also, communication time is minimized when $\beta \max_i |R_i|$ and $\max_j |S_j|$ are minimal. For fixed $\alpha, \beta$, since $\{R_i\}_{i=1}^{\alpha}$ and $\{S_j\}_{j=1}^{\beta}$ partition $[N]$, then $\max_i |R_i| \geq \lceil N/\alpha \rceil$ and $\max_j |S_j| \geq \lceil N/\beta \rceil$. Equality holds when all $R_i$, and all $S_j$, are around the same size. Thus, $(c, \delta)$-sharding has optimal communication time.

### 7.3.2. PERFORMANCE EXPERIMENTS

We now evaluate the real-world performance of Cascade through the distributed computing framework Ray (Moritz et al., 2018). We run our experiments on Paperspace machines with 16 vCPU and 64GB RAM, with the CPU model being Intel Xeon Gold 6226R CPU @ 2.90GHz. All machines are colocated in the same region with an average bandwidth of 2 Gbps and latency of 0.38 ms. We repeat a single forward pass on Bert-Base and Bert-Large with a 128-token prompt a total of 100 times.

We benchmark against two recent SMPC schemes for LLM inference, MPCFormer (Li et al., 2023a) and Puma (Dong et al., 2023b). For MPCFormer, we modify the Crypten implementation to use public rather than private weights, to match our open-weights setting. Puma data is taken from Dong et al. (2023b), as it is built on SPU with its own set of optimizations. Our results are shown in Table 6.

We first compare Cascade for $\alpha = 1$ without Ray against vanilla inference, to estimate protocol overhead. The mean runtime is 109ms vs. 91ms for vanilla inference for Bert-Base, and 320ms vs. 273ms for Bert-Large. Profiling shows this minor increase is due to the attention-score compilation step discussed in Appendix N. Nevertheless, the mean runtime is within the 95% confidence interval of vanilla inference in both cases.

Next, we compare the performance of Cascade with $\alpha = 1$ and using Ray. As seen above, this is slower than not using Ray by around a factor of $3\times$. This slowdown can be attributed to framework-specific overhead, such as serialization. In other words, Cascade is so efficient that the distributed-compute framework overhead now constitutes a significant proportion of its slowdown from vanilla inference, rather than protocol-specific overhead.

*Table 6.* Total runtime means and 95% confidence intervals in seconds, for a single 128-token prompt forward pass on Bert-Base and Bert-Large for MPCFormer, Puma, and some Cascade settings.

| Scheme | Bert-Base (s) | Bert-Large (s) |
|---|---|---|
| MPCFormer | 55.320 | 141.222 |
| Puma | 33.913 | 73.720 |
| Cascade$_{\alpha=1,\text{no Ray}}$ | 0.11 [0.10, 0.13] | 0.32 [0.23,1.07] |
| Cascade$_{\alpha=1}$ | 0.32 [0.31, 0.36] | 1.01 [0.97, 1.09] |
| Cascade$_{\alpha=4}$ | 0.59 [0.51,0.69] | 1.57 [1.44, 1.73] |
| Cascade$_{\alpha=8}$ | 0.74 [0.62, 0.96] | 1.58 [1.27, 1.97] |
| *Vanilla* | 0.09 [0.08,0.12] | 0.27 [0.20, 0.99] |

We further benchmark Cascade with $\alpha = 4, 8$, using a $(c, \delta)$-sharding scheme with $c = 1$ and no $m$-splits. We use clusters of 6 and 18 machines for $\alpha = 4$ and 8, with 4 cores per node. Performance in these cases is slower than for $\alpha = 1$, due to communication overhead. Still, Cascade is $90\times$ faster than MPCFormer and $45\times$ faster than Puma for Bert-Large, even in its slowest setting. In Appendix R, we further test Cascade runtime across various model sizes, and find that it scales well to larger models.

*Table 7.* Communicated bytes for a 128-token prompt forward pass on Bert-Base and Bert-Large for MPCFormer, Puma, and Cascade.

| Scheme | Bert-Base (GB) | Bert-Large (GB) |
|---|---|---|
| MPCFormer | 12.089 | 32.577 |
| Puma | 10.773 | 27.246 |
| Cascade$_{\alpha=1}$ | 0.009 | 0.025 |
| Cascade$_{\alpha=4}$ | 0.038 | 0.101 |
| Cascade$_{\alpha=8}$ | 0.076 | 0.203 |

Finally, we compare total communicated bytes for Cascade versus MPCFormer and Puma in Table 7 above. Even in the most expensive setting, Cascade is $160\times$ more efficient than MPCFormer and $140\times$ more efficient than Puma. Thus, with significant improvements in computation and communication cost, Cascade offers a new paradigm in the trade-off between scalability and security.

## 8. Conclusion & Future Work

We identify a new attack for decoding LLM hidden states into their original user text in the increasingly important open-weights setting. This nearly perfectly decodes even permuted hidden states, effectively invalidating the security of some existing MPC schemes. We also introduce a novel multi-party scheme, Cascade, that leverages token sharding to defend against our attack and existing attacks in the literature. Future work could investigate the choice of $\epsilon$ threshold in our attack, or examine the security of alternative sharding strategies in Cascade.

## Impact Statement

This paper presents work whose goal is to advance the field of Machine Learning, and particularly, of privacy-preserving inference of LLMs. Our work may lead to the enactment of more secure and scalable methods of private inference in the open-weights setting. There are many potential societal consequences of this, none of which we feel must be specifically highlighted here.

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

## A. Vocab-Matching Attack Optimizations

Although the cost of the attack outlined in Section 4 is linear in $V$, the size of vocabularies can be quite large in practice. For example, Gemma-2-2B-IT has a vocabulary size of 256000. Therefore we seek to optimize this by introducing a *proposal model*. The purpose of the proposal model is to provide a suggested ordering over the vocabulary, rather than iterate through it in an arbitrary order. It does so by taking in the token sequence that has been partially decoded so far and producing the next-token logits. We then search through the next-token logits in decreasing order of probability. In practice, we find that this modification reduces the expected number of tokens searched through at each step from $V/2$ to approximately 100, thus representing a constant factor speedup of more than $1000\times$.

Moreover, we implement a novel variation of key-value-caching (KV-caching) to reduce the computational time of our attack. Note that at the $n$th stage of the decoding, we are performing a $V$-batched forward pass on $[\widehat{x}_1, \widehat{x}_2, ..., \widehat{x}_{n-1}, v]$ over $v \in \mathcal{V}$, where $\widehat{x}_1, \widehat{x}_2, \ldots, \widehat{x}_{n-1}$ are the tokens that we have already decoded. As this forward pass needs to be repeated many times for different $v$ but the same $\widehat{x}_i$, we cache the keys and values associated to the $\widehat{x}_i$ and reuse them across all forward passes. This is different from standard KV-caching, which stores the keys and values for generation over a single sequence: here, we reuse keys and values across many sequences. In practice, this optimized caching gives a significant speedup in vocab-matching: across 10 evaluation prompts from FineWeb, the average caching speedup was around $20\times$, with speedups for all prompts in the range 15-30$\times$.

## B. Scalability of Attack

To assess the scalability of our attack with respect to model size, we conducted additional experiments across a range of model scales, from 1 billion to 27 billion parameters. Table 8 summarizes the results.

*Table 8.* Average attack time (in seconds) over 10 decodings for various model sizes.

| Model Name | Model Size (Parameters) | Vocabulary Size | Average Attack Time (s) |
|---|---|---|---|
| Llama-3.2-1B-Instruct | 1B | 128,256 | 49 |
| Gemma-2-2B-IT | 2B | 256,000 | 124 |
| Llama-3.1-8B-Instruct | 8B | 128,256 | 69 |
| Gemma-2-27B-IT ($\epsilon = 30$) | 27B | 256,000 | 304 |
| Gemma-2-27B-IT ($\epsilon = 40$) | 27B | 256,000 | 124 |

The attack time remains practical across all evaluated model sizes, typically on the order of minutes for perfect decoding of length 100 prompts. We observe that the computational cost is primarily a function of the vocabulary size and the choice of $\epsilon$, rather than the total number of model parameters. Specifically, models with larger vocabularies (e.g., 256,000 tokens) exhibit proportionally longer attack times compared to models with smaller vocabularies (e.g., 128,256 tokens), regardless of parameter count. While a poorly chosen $\epsilon$ leads to longer runtimes, it does not fundamentally impede the attack. These results demonstrate that the attack scales favorably to larger models, including recent LLMs with tens of billions of parameters.

## C. Vocab-Matching Attack Walkthrough and Gap Generalization

Here, we provide an in-depth walkthrough of Algorithm 5 on an example prompt, where the adversary receives layer $L$ hidden states $h_1, \ldots, h_N \in \mathbb{R}^d$ and attempts to recover the $N$-token input prompt. Then, we walk through a variant where the adversary only receives some of these $N$ hidden states but can perform a search over *multiple* unknown tokens.

**Vanilla Attack** Suppose a user provides the following prompt to an LLM inference provider hosting Gemma-2-2B-IT: "What is currently the most populated city in Spain?". After tokenization, this might be represented as the list [BOS, "What", "is", "currently", "the", "most", "populated", "city", "in", "Spain", "?", EOS], with BOS and EOS denoting special beginning or end of sequence tokens. These 12 tokens have layer 0 embeddings $e_1, \ldots, e_{12} \in \mathbb{R}^d$, and these give layer $L$ embeddings $h_1, \ldots, h_{12} \in \mathbb{R}^d$. Denoting the stacked decoder layers in Gemma-2-2B-IT as $\phi_1, \ldots, \phi_f$, which are all unidirectional and assumed to be deterministic (for now), one sees for $\phi_{\leq L} = \phi_L \circ \ldots \circ \phi_2 \circ \phi_1$ that

$$\phi_{\leq L}(e_1) = [h_1] \in \mathbb{R}^{1 \times d}, \ \phi_{\leq L}(e_1, e_2) = [h_1, h_2] \in \mathbb{R}^{2 \times d}, \ \ldots, \ \phi_{\leq L}(e_1, \ldots, e_{12}) = [h_1, \ldots, h_{12}] \in \mathbb{R}^{12 \times d}.$$

We first describe how the attacks works without the $\epsilon$-threshold, i.e. setting $\epsilon < 0$, and then describe how $\epsilon$ affects this.

First, the provider takes the Gemma-2-2B-IT vocabulary $\mathcal{V}$ with size $V = 256000$. For each token $v \in \mathcal{V}$ with embedding $e_v \in \mathbb{R}^d$, they compute $\phi_{\leq L}(e_v) \in \mathbb{R}^{1 \times d}$, and then set $\widehat{e}_1$ as some $e_v$ which minimizes the L1 error $\|\phi_{\leq L}(e_v)_1 - h_1\|_1$. Without non-determinism, $\phi_{\leq L}(e_1) = h_1$, so assuming no collisions (no other $\phi_{\leq L}(e_v) = h_1$), we will have $\widehat{e}_1 = e_1$. Next, they compute $\phi_{\leq L}(\widehat{e}_1, e_v) \in \mathbb{R}^{2 \times d}$ for each $v \in \mathcal{V}$, and set $\widehat{e}_2$ as some $e_v$ which minimizes $\|\phi_{\leq L}(\widehat{e}_1, e_v)_2 - h_1\|_1$. Again, without non-determinism and collisions, we have $\widehat{e}_2 = e_2$. This continues until finally, they compute $\phi_{\leq L}(\widehat{e}_1, \ldots, \widehat{e}_{11}, e_v) \in \mathbb{R}^{11 \times d}$, and set $\widehat{e}_{12}$ as some $e_v$ which minimizes $\|\phi_{\leq L}(\widehat{e}_1, \ldots, \widehat{e}_{11}, e_v)_{12} - h_{12}\|_1$, which will be $e_{12}$. Now, the provider has true input embeddings $[\widehat{e}_1, \ldots, \widehat{e}_{12}] = [e_1, \ldots, e_{12}]$, which can be reversed into the 12 input tokens via the lookup table.

Here, non-determinism and non-collidingness are necessary for *provable* success of the attack, as otherwise we may not have each $\widehat{e}_i = e_i$. Furthermore, a total of $12V$ (around 3 million) forward passes are performed in the worst case, with the exact count depending on the order of iteration in $\mathcal{V}$. As this is expensive, we introduce $\epsilon$-thresholding: instead of fully iterating through $\mathcal{V}$, we stop once we find an $e_v$ with associated L1 error $\leq \epsilon$. However, with a random order of iteration on $\mathcal{V}$, we might not expect significant speedups from this, since the correct token (or other tokens with error $\leq \epsilon$) might appear late. Thus, we use a proposal model: this conditions each the order of iteration through $\mathcal{V}$ on already deciphered tokens, allowing the $\epsilon$-stopping condition to be met earlier. The optimizations from $\epsilon$ and the proposal model make the forward pass count much less than 3 million in practice, but potentially at the cost of accuracy: if the proposal model suggests an incorrect token with L1 error $\leq \epsilon$ earlier than the correct token, the attack may not be able to decode any future tokens.

**Generalized Attack**   Now, we discuss a generalization of the vanilla attack above, shown in Algorithm 3. Here, we discuss how it applies to the example from the last section, with the same notation. Since we do not implement this attack, and consider it primarily as a theoretical threat, we forgo details such as the proposal model and $\epsilon$, and we assume non-determinism. The generalized attack pertains to settings where the provider does not receive *all* $N$ hidden states. For instance, suppose they only receive $h_2, h_5, h_{10}$. Can they still recover all input tokens?

The answer is negative. For instance, none of these 3 hidden states are functions of $e_{11}, e_{12}$, so one cannot fully[4] guarantee recovery tokens 11 and 12. Nevertheless, our generalized attack, with sufficiently large computation cost, allows recovery of all tokens that the last hidden state $h_{10}$ depends on, meaning the first 10 tokens.

First, since the provider does not know $h_1$, they cannot carry out the first step of the vanilla attack to decipher $e_1$. However, from $h_3$, they can obtain something stronger: $e_1, e_2, e_3$. This is because $[h_1, h_2, h_3] = \phi_{\leq L}(e_1, e_2, e_3)$, so assuming non-collidingness[5] of the $V^3$ possible 3-token forward passes $\{\phi_{\leq L}(e_{v_1}, e_{v_2}, e_{v_3})\}_{v_1, v_2, v_3 \in \mathcal{V}}$, the provider can recover $e_1, e_2, e_3$ in at most $V^3$ forward passes. Next, from $[h_1, \ldots, h_5] = \phi_{\leq L}(e_1, \ldots, e_5)$ and non-collidingness of the $V^2$ possible 2-token forward passes $\{\phi_{\leq L}(e_1, e_2, e_3, e_{v_4}, e_{v_5})\}_{v_4, v_5 \in \mathcal{V}}$, the provider can recover $e_4, e_5$ in at most $V^2$ forward passes. Finally, from $[h_1, \ldots, h_{10}] = \phi_{\leq L}(e_1, \ldots, e_{10})$ and non-collidingness of the $V^5$ possible 5-token forward passes $\{\phi_{\leq L}(e_1, \ldots, e_5, e_{v_6}, \ldots, e_{v_{10}})\}_{v_6, \ldots, v_{10} \in \mathcal{V}}$, the provider can recover $e_6, \ldots, e_{10}$ in at most $V^5$ forward passes.

Essentially, the provider has recovered the first 10 tokens in $\leq V^3 + V^2 + V^5$ forward passes, where exponents $3, 2, 5$ correspond to gaps between indices of known hidden states. In general, if the provider gets $h_{i_1}, \ldots, h_{i_k}$ with $1 \leq i_1 < \ldots < i_k \leq N$, then they can carry out this attack to reverse the first $i_k$ tokens in $\leq V^{i_1} + V^{i_2 - i_1} + \ldots + V^{i_k - i_{k-1}}$ forward passes. Although this attack seems quite powerful, the upper bound on forward passes can be quite large, so it is not always practical. For instance, the dominating term in $V^3 + V^2 + V^5$ is $V^5 \approx 10^{27}$, so even if only a fraction of the upper bound cost is achieved, it will time out. In general, as $k \leq N = o(V)$ in practice, the cost bound $V^{i_1} + V^{i_2 - i_1} + \ldots + V^{i_k - i_{k-1}}$ is dominated by the maximum-gap term $V^g$ with $g = \max_j (i_{j+1} - i_j)$. Thus, whenever $g$ is large enough so that the adversary *cannot* perform $V^g$ forward passes, we would expect the attack to time out, even with proposal models and other optimizations. The minimum value of $g$ satisfying this is precisely the **vocab-matching threshold** $\rho$ from Definition 7.2.

Finally, we remark that the the generalized attack requires some stronger assumptions than the vanilla attack to succeed. For instance, in the vanilla attack, we only need non-collidingness across $V$ forward passes at a time. But in our example, we need it across up to $V^5$ forward passes, and in general, we need it across up to $V^g$ forward passes. One interesting question along this line is to find (if it even exists) the minimum token gap where there are significant collisions at some layer, for various LLMs. With many gaps near this threshold, defenses like quantization and noise could prove more effective than they do in Appendix F. We leave such work as an interesting future direction.

---

[4]One could infer these from all previous tokens for certain prompts, so here, we mean without assuming a prior on text. For instance, in the example prompt, "?" and EOS could be inferred from the rest of the prompt.

[5]Actually, since we are matching only the last row to $h_3$, all that is needed is non-collidingness of the last row.

**Algorithm 3** Generalized Vocabulary-Matching Attack

---

**input** Model $M$, indices $1 \leq i_1 < i_2 < \ldots < i_k \leq N$, layer $l$ hidden states $[h_{i_1}, h_{i_2}, \ldots, h_{i_k}]$, vocabulary $\mathcal{V}$

**output** Nearly decoded token sequence $\widehat{\boldsymbol{x}} = [\widehat{x}_1, \widehat{x}_2, \ldots, \widehat{x}_{i_k - 1}, \widehat{x}_{i_k}]$

1: Initialize empty sequence $\widehat{\boldsymbol{x}} \leftarrow []$
2: $i_0 \leftarrow 0$
3: **for** $j = 0$ to $k - 1$ **do**
4:     gap $\leftarrow i_{j+1} - i_j$
5:     min_dist $\leftarrow \infty$
6:     best_match $\leftarrow$ None
7:     **for** $v_1, v_2, \ldots, v_{\text{gap}} \in \mathcal{V}$ **do**
8:         $g \leftarrow M_{\leq l}([\widehat{\boldsymbol{x}}, v_1, v_2, \ldots, v_{\text{gap}}])$ {Forward pass up to layer $l$}
9:         dist $\leftarrow \|g - h_{i_{j+1}}\|_1$ {Calculate L1 distance}
10:         **if** dist < min_dist **then**
11:             min_dist $\leftarrow$ dist
12:             best_match $\leftarrow v$
13:         **end if**
14:     **end for**
15: **end for**
16: **return** $\widehat{\boldsymbol{x}}$

---

## D. Optimal $\epsilon$ for Decoding

We report the full set of optimal $\epsilon$ thresholds in decoding, for each permutation type below. We observe that generally, the optimal $\epsilon$ increases in later layers across all permutation types – which may be due to the effect of the reducible and irreducible noise we mention in Section 4 taking up a larger subspace volume as it propagates to deeper layers. We also observe that Llama tends to have much lower $\epsilon$ values in general.

There is also an interesting *layerwise* distinction between Gemma and Llama: the optimal $\epsilon$ for the last hidden layer (26) in Gemma decreases by nearly $2\times$ outside of the no permutation case. But the opposite is the case for Llama: it increases by more than $2\times$ at the last layer (32) for all but the no permutation case. Both Gemma and Llama have slightly decreased $\epsilon$ at the last layer in the no permutation case. Investigating the reason for decreasing versus increasing $\epsilon$-ball collisions in the last few layers, based on distinctions in the architecture or weights of models like Gemma and Llama, and the type of permutation applied, is an interesting direction for future work on vocab-matching.

*Table 9.* Optimal $\epsilon$ thresholds for hidden state reversal without permutation, over various Gemma-2-2B-IT and Llama-3.1-8B-Instruct layers.

| Layer | Gemma | Llama |
|:-----:|:-----:|:-----:|
| 1 | 22.0 | 0.6 |
| 6 | 70.0 | 7.1 |
| 11 | 204.0 | 18.3 |
| 16 | 293.0 | 29.0 |
| 21 | 400.0 | 76.0 |
| 26 | 318.0 | 156.0 |
| 32 | — | 150.0 |

*Table 10.* Optimal $\epsilon$ thresholds for hidden state reversal with sequence dimension permutation, over various Gemma-2-2B-IT and Llama-3.1-8B-Instruct layers.

| Layer | Gemma | Llama |
|-------|-------|-------|
| 1 | 12.8 | 1.4 |
| 6 | 72.6 | 3.3 |
| 11 | 229.0 | 7.4 |
| 16 | 301.0 | 7.4 |
| 21 | 385.0 | 26.6 |
| 26 | 220.0 | 29.6 |
| 32 | — | 105.0 |

*Table 11.* Optimal $\epsilon$ thresholds for hidden state reversal with hidden dimension permutation, over various Gemma-2-2B-IT and Llama-3.1-8B-Instruct layers.

| Layer | Gemma | Llama |
|-------|-------|-------|
| 1 | 12.5 | 0.5 |
| 6 | 25.0 | 3.5 |
| 11 | 45.0 | 3.7 |
| 16 | 73.0 | 5.2 |
| 21 | 118.0 | 6.3 |
| 26 | 61.0 | 9.8 |
| 32 | — | 30.0 |

*Table 12.* Optimal $\epsilon$ thresholds for hidden state reversal with factorized-2D permutation, over various Gemma-2-2B-IT and Llama-3.1-8B-Instruct layers.

| Layer | Gemma | Llama |
|-------|-------|-------|
| 1 | 21.0 | 0.3 |
| 6 | 26.0 | 3.0 |
| 11 | 47.0 | 9.0 |
| 16 | 69.0 | 9.0 |
| 21 | 118.0 | 14.0 |
| 26 | 51.0 | 14.0 |
| 32 | — | 45.0 |

## E. Vocab-Matching in Permuted Settings

In Section 5, we provided a high-level description of how to modify the attack in Section 4 to permuted setting. Below, we provide exact psuedocode for the attack in sequence dimension, hidden dimension, and factorized-2D permutation settings.

---

**Algorithm 4** Attack on Sequence Dimension Permuted LLM Hidden States

---

**input** Model $M$, permuted layer $l$ hidden states $\boldsymbol{h} = [h_{\sigma(1)}, h_{\sigma(2)}, ..., h_{\sigma(N)}]$, vocabulary $\mathcal{V}$, proposal model $P$, L1-threshold $\epsilon$

**output** Decoded token sequence $\widehat{\boldsymbol{x}} = [\widehat{x}_1, \widehat{x}_2, \ldots, \widehat{x}_N]$

1: Initialize empty sequence $\widehat{\boldsymbol{x}} \leftarrow []$
2: Initialize set of remaining hidden states $\mathcal{H} \leftarrow \{h_{\sigma(1)}, h_{\sigma(2)}, ..., h_{\sigma(N)}\}$
3: **for** $i = 1$ to $N$ **do**
4:    $\mathcal{V}_{\text{ordered}} \leftarrow \text{argsort}(P([\widehat{\boldsymbol{x}}, v]|\widehat{\boldsymbol{x}}))$ {Get ordered vocabulary from proposal model}
5:    min_dist $\leftarrow \infty$
6:    best_match $\leftarrow$ None
7:    **for** $v \in \mathcal{V}_{\text{ordered}}$ **do**
8:       $g \leftarrow M_{\leq l}([\widehat{\boldsymbol{x}}, v])$ {Forward pass up to layer $l$}
9:       **for** $h \in \mathcal{H}$ **do**
10:          dist $\leftarrow \|g - h\|\|_1$ {Calculate L1 distance}
11:          **if** dist $<$ min_dist **then**
12:             min_dist $\leftarrow$ dist
13:             best_match $\leftarrow v$
14:             best_h $\leftarrow h$
15:          **end if**
16:          **if** dist $< \epsilon$ **then**
17:             $\widehat{x}_i \leftarrow v$
18:             Remove $h$ from $\mathcal{H}$
19:             break
20:          **end if**
21:       **end for**
22:    **end for**
23:    **if** min_dist $\geq \epsilon$ **then**
24:       $\widehat{x}_i \leftarrow$ best_match
25:       Remove best_h from $\mathcal{H}$
26:    **end if**
27: **end for**
28: **return** $\widehat{\boldsymbol{x}}$

---

---

**Algorithm 5** Attack on Hidden Dimension Permuted LLM Hidden States

---

**input** Model $M$, layer $l$ permuted hidden states $\boldsymbol{h} = [\pi_1(h_1), \pi_2(h_2), ..., \pi_N(h_N)]$, vocabulary $\mathcal{V}$, proposal model $P$,
    L1-threshold $\epsilon$

**output** Decoded token sequence $\widehat{\boldsymbol{x}} = [\widehat{x}_1, \widehat{x}_2, \ldots, \widehat{x}_N]$

 1: Initialize empty sequence $\widehat{\boldsymbol{x}} \leftarrow []$
 2: **for** $i = 1$ to $N$ **do**
 3:     $\mathcal{V}_{\text{ordered}} \leftarrow \text{argsort}(P([\widehat{\boldsymbol{x}}, v]|\widehat{\boldsymbol{x}}))$ {Get ordered vocabulary from proposal model}
 4:     min_dist $\leftarrow \infty$
 5:     best_match $\leftarrow$ None
 6:     **for** $v \in \mathcal{V}_{\text{ordered}}$ **do**
 7:         $g \leftarrow M_{\leq l}([\widehat{\boldsymbol{x}}, v])$ {Forward pass up to layer $l$}
 8:         dist $\leftarrow \|\text{sort}(g) - \text{sort}(\pi_i(h_i))|\|_1$ {Calculate L1 distance of sorted vectors}
 9:         **if** dist $<$ min_dist **then**
10:            min_dist $\leftarrow$ dist
11:            best_match $\leftarrow v$
12:         **end if**
13:         **if** dist $< \epsilon$ **then**
14:            $\widehat{x}_i \leftarrow v$
15:            break
16:         **end if**
17:     **end for**
18:     **if** min_dist $\geq \epsilon$ **then**
19:         $\widehat{x}_i \leftarrow$ best_match
20:     **end if**
21: **end for**
22: **return** $\widehat{\boldsymbol{x}}$

---

---

**Algorithm 6** Attack on Factorized-2D Permuted LLM Hidden States

---

**input** Model $M$, permuted layer $l$ hidden states $\boldsymbol{h} = [\pi_1(h_{\sigma(1)}), \pi_2(h_{\sigma(2)}), ..., \pi_N(h_{\sigma(N)})]$, vocabulary $\mathcal{V}$, proposal model $P$, L1-threshold $\epsilon$

**output** Decoded token sequence $\widehat{\boldsymbol{x}} = [\widehat{x}_1, \widehat{x}_2, \ldots, \widehat{x}_N]$

1: Initialize empty sequence $\widehat{\boldsymbol{x}} \leftarrow []$
2: Initialize set of remaining hidden states $\mathcal{H} \leftarrow \{\pi_1(h_{\sigma(1)}), \pi_2(h_{\sigma(2)}), ..., \pi_N(h_{\sigma(N)})\}$
3: **for** $i = 1$ to $N$ **do**
4:     $\mathcal{V}_{\text{ordered}} \leftarrow \text{argsort}(P([\widehat{\boldsymbol{x}}, v]|\widehat{\boldsymbol{x}}))$ {Get ordered vocabulary from proposal model}
5:     $\text{min\_dist} \leftarrow \infty$
6:     $\text{best\_match} \leftarrow \text{None}$
7:     **for** $v \in \mathcal{V}_{\text{ordered}}$ **do**
8:         $g \leftarrow M_{\leq l}([\widehat{\boldsymbol{x}}, v])$ {Forward pass up to layer $l$}
9:         **for** $h \in \mathcal{H}$ **do**
10:           $\text{dist} \leftarrow \|\text{sort}(g) - \text{sort}(h)\|_1$ {Calculate L1 distance of sorted vectors}
11:           **if** $\text{dist} < \text{min\_dist}$ **then**
12:             $\text{min\_dist} \leftarrow \text{dist}$
13:             $\text{best\_match} \leftarrow v$
14:             $\text{best\_h} \leftarrow h$
15:           **end if**
16:           **if** $\text{dist} < \epsilon$ **then**
17:             $\widehat{x}_i \leftarrow v$
18:             Remove $h$ from $\mathcal{H}$
19:             break
20:           **end if**
21:         **end for**
22:     **end for**
23:     **if** $\text{min\_dist} \geq \epsilon$ **then**
24:         $\widehat{x}_i \leftarrow \text{best\_match}$
25:         Remove $\text{best\_h}$ from $\mathcal{H}$
26:     **end if**
27: **end for**
28: **return** $\widehat{\boldsymbol{x}}$

---

# F. Noised & Quantized Hidden State Reversal

As mentioned in Section 6, we now examine the efficacy of our attack on methods of defense that modify the hidden states directly – such as by adding noise, or by quantizing the model to a lower precision. We investigate the following methods of modification to the permuted LLM hidden states:

- Adding diagonal Gaussian noise with mean 0 and standard deviation $\sigma$ to each hidden dimension in the input embeddings, as proposed in Morris et al. (2023a).

- Inserting a randomly generated embedding as a prefix to the original sequence. This has the effect of modifying the subsequent hidden states via self-attention.

- Quantization of the model.

Clearly, with a sufficiently high degree of noise, decoding can be made impossible. However, high noise will also likely disrupt LLM performance. Therefore, the crux of any such defense is based on the delicate balancing act of ensuring security against our attack, whilst still maintaining downstream model performance. We further consider the combination of both permutation as well as the above noising methods.

## F.1. Experiments

We apply each of the above noising methods on Gemma-2-2B-IT. For diagonal Gaussian noise, we test with $\sigma = 0.1, 0.01$. For the random embedding prefix, we generate the embedding from a Gaussian with means and standard deviations of each hidden dimension set to the average over the token vocabulary $\mathcal{V}$. For quantization, we test with reduction of the model from its original 16-bit to 8-bit and 4-bit, using the bitsandbytes library (BitsAndBytes, 2025). We apply each of the above methods to all the permutation types described in Section 5, as well as the unpermuted hidden states. Our choice of dataset, number of evaluation samples, and method of choosing $\epsilon$ is the same as in Section 4.4. As perfect decoding is less commonly achieved with the addition of noise, we now report the ROUGE-L score between the decoded reconstruction and the original prompt to measure decoding quality. We conduct testing again over layers 1, 6, 11, 16, 21 and 26, but report only the highest ROUGE-L, as this can be considered the weakest attack point.

To measure the downstream impact of the noising methods, we utilize LiveBench (White et al., 2024), a benchmark that tests across multiple different components of LLM performance, such as language, reasoning and math. Our results are given in Table 13 below. A full breakdown of the LiveBench scores by category and the ROUGE-L scores by layer of each of the above methods and permutation types is given in Appendix S.

*Table 13.* ROUGE-L reconstruction scores across 1000 evaluation samples for various noising methods and permutation types on Gemma-2-2B-IT. The 'Downstream Performance' column is the normalized score on LiveBench (White et al., 2024), a benchmark that tests broad components of LLM performance such as math, reasoning and language. Note that LiveBench scores carry some variability, and so the baseline, Gaussian with standard deviation 0.01, and random embedding prefix methods are all within noise in performance.

| Method | Unpermuted | Sequence Perm | Hidden Perm | Factorized-2D | Downstream Performance |
|---|---|---|---|---|---|
| Baseline (no noise) | 1.00 | 1.00 | 1.00 | 1.00 | 100.0% |
| Gaussian, $\sigma = 0.01$ | 0.93 | 0.07 | 0.07 | 0.07 | 101.4% |
| Gaussian, $\sigma = 0.1$ | 0.91 | 0.01 | 0.01 | 0.01 | 5.8% |
| Random emb. prefix | 0.93 | 0.17 | 0.19 | 0.19 | 102.9% |
| 8-bit quantization | 0.89 | 0.86 | 0.75 | 0.73 | 97.6% |
| 4-bit quantization | 0.88 | 0.84 | 0.83 | 0.71 | 92.2% |

We see that unpermuted hidden states are still highly decodeable via our attack under all methods tested – the ROUGE-L scores are above 0.8 in all cases, indicating significant similarity with the original text. Remarkably, even 4-bit quantization is not sufficient to introduce enough collisions to significantly mitigate our attack. We find that the combination of permutation and Gaussian noise with standard deviation 0.01 appears largely secure, with ROUGE-L scores below 0.1, and maintains downstream performance, and thus may represent a potential solution to the insecurity of STIP and Centaur. However, this

result is only necessary for security, and not sufficient; it is possible that extensions of our attack family can succeed even in this setting. We leave further investigation of this to future work.

## G. Cascade Scheme Details

Now, we describe the pre-pass, attention-pass, and post-pass components of Cascade, which were introduced in Section 7.1. All shard-specific slicing and concatenation operations are initialized in the node setup. Furthermore, we assume that all AttnNodes wait until all CompNodes finish the pre-pass to do the attention-pass, and all CompNodes wait until all AttnNodes finish the attention-pass to do the post-pass.

**Pre-pass**    At layer $l$, each CompNode$_i$ starts with the $R_i-$sharded hidden states $\boldsymbol{h}^{R_i}$, and applies layer normalization if necessary. Then, it $Q, K, V$-projects these to get the $R_i-$sharded query, key and value states $\boldsymbol{q}^{R_i}, \boldsymbol{k}^{R_i}, \boldsymbol{v}^{R_i}$. CompNode$_i$ then applies rotary or positional embedding to $\boldsymbol{q}^{R_i}, \boldsymbol{k}^{R_i}$, using sharded positional embeddings $\boldsymbol{p}^{R_i}$ (the node can generate these upon setup to avoid any communication overhead, since it only depends on its index set $R_i$), and returns all of $\boldsymbol{q}^{R_i}, \boldsymbol{k}^{R_i}, \boldsymbol{v}^{R_i}$, as described in Algorithm 7.

**Attention-pass**    After the pre-pass, each AttnNode$_{jk}$ receives shards $\boldsymbol{q}^{R_i S_j}, \boldsymbol{k}^{R_i S_k}, \boldsymbol{v}^{R_i S_k}$ from CompNode$_i$. By concatenating their rows over $1 \leq i \leq \alpha$, AttnNode$_{jk}$ obtains $\boldsymbol{q}^{S_j}, \boldsymbol{k}^{S_k}, \boldsymbol{v}^{S_k}$. For example, for $\boldsymbol{q}^{S_j}$, concatenation is in the order in which one concatenates elements of *sorted* $R_i \cap S_j$ over $1 \leq i \leq \alpha$ to get *sorted* $S_j$. Then, AttnNode$_{jk}$ computes $\boldsymbol{a}^{S_j S_k} = \boldsymbol{q}^{S_j}(\boldsymbol{k}^{S_k})^T + \boldsymbol{s}^{S_j S_k}$, where matrix multiplication is per-head and we broadcast $H_{KV}$ to $H$. For the post-pass, AttnNode$_{jk}$ also stores row-wise maximums and subtract-max expsums $\boldsymbol{m}^{S_j S_k}, \boldsymbol{e}^{S_j S_k}$. Finally, AttnNode$_{jk}$ takes the row-wise softmax and performs value multiplication to get $\boldsymbol{u}^{S_j S_k} = \text{softmax}(\boldsymbol{a}^{S_j S_k})\boldsymbol{v}^{S_k}$. All of $\boldsymbol{m}^{S_j S_k}, \boldsymbol{e}^{S_j S_k}, \boldsymbol{u}^{S_j S_k}$ are returned, as in Algorithm 8.

**Post-pass**    Finally, after the attention-pass, each CompNode$_i$ receives $\boldsymbol{m}^{(R_i \cap S_j)S_k}, \boldsymbol{e}^{(R_i \cap S_j)S_k}, \boldsymbol{u}^{(R_i \cap S_j)S_k}$ from each AttnNode$_{jk}$, which are slices of its attention-pass outputs $\boldsymbol{m}^{S_j S_k}, \boldsymbol{e}^{S_j S_k}, \boldsymbol{u}^{S_j S_k}$ along the second-to-last dimensions. Then, for each fixed $1 \leq k \leq \beta$, CompNode$_i$ concatenates the rows of $\boldsymbol{m}^{(R_i \cap S_j)S_k}, \boldsymbol{e}^{(R_i \cap S_j)S_k}, \boldsymbol{u}^{(R_i \cap S_j)S_k}$ over all $1 \leq j \leq \beta$ to obtain $\boldsymbol{m}^{R_i S_k}, \boldsymbol{e}^{R_i S_k}, \boldsymbol{u}^{R_i S_k}$. Next, CompNode$_i$ aims to combine these results $\boldsymbol{m}^{R_i S_k}, \boldsymbol{e}^{R_i S_k}, \boldsymbol{u}^{R_i S_k}$ over $1 \leq k \leq \beta$ into the $R_i-$sharded (pre $O$-proj) output of attention. Using slicing notation, treating matrix multiplication as per-head, and broadcasting $H_{KV}$ to $H$, this is

$$\text{softmax}(\boldsymbol{a})[:, R_i, :]\boldsymbol{v} = \sum_{k=1}^{\beta} \text{softmax}(\boldsymbol{a})[:, R_i, S_k]\boldsymbol{v}[:, :, S_k] \in \mathbb{R}^{H \times |R_i| \times d}$$

by blocked matrix multiplication. The terms in the summation are not known to CompNode$_i$, since slicing here is performed post-softmax. To correct for this, observe that for a row vector $\boldsymbol{x} \in \mathbb{R}^N$ and any $1 \leq k \leq \beta$, again with slicing notation,

$$\text{softmax}(\boldsymbol{x})[S_k] = \frac{\text{expsum}(\boldsymbol{x}[S_k])}{\sum_l \text{expsum}(\boldsymbol{x}[S_l])} \cdot \text{softmax}(\boldsymbol{x}[S_k]) \in \mathbb{R}^{|S_k|}.$$

Thus, the above summation can be simplified as follows, with $\odot$ denoting elementwise multiplication:

$$\begin{aligned}
\text{softmax}(\boldsymbol{a})[:, R_i, :]\boldsymbol{v} &= \frac{\sum_k \text{expsum}(\boldsymbol{a}[:, R_i, S_k]) \odot (\text{softmax}(\boldsymbol{a}[:, R_i, S_k])\boldsymbol{v}[:, S_k])}{\sum_k \text{expsum}(\boldsymbol{a}[:, R_i, S_k])} \\
&= \frac{\sum_k \text{expsum}(\boldsymbol{a}^{R_i S_k}) \odot (\text{softmax}(\boldsymbol{a}^{R_i S_k})\boldsymbol{v}^{S_k})}{\sum_k \text{expsum}(\boldsymbol{a}^{R_i S_k})} \\
&= \frac{\sum_k \exp(\boldsymbol{m}^{R_i S_k}) \odot \text{expsum}(\boldsymbol{a}^{R_i S_k} - \boldsymbol{m}^{R_i S_k}) \odot (\text{softmax}(\boldsymbol{a}^{R_i S_k})\boldsymbol{v}^{S_k})}{\sum_k \exp(\boldsymbol{m}^{R_i S_k}) \odot \text{expsum}(\boldsymbol{a}^{R_i S_k} - \boldsymbol{m}^{R_i S_k})} \\
&= \frac{\sum_k \exp(\boldsymbol{m}^{R_i S_k}) \odot \boldsymbol{e}^{R_i S_k} \odot \boldsymbol{u}^{R_i S_k}}{\sum_k \exp(\boldsymbol{m}^{R_i S_k}) \odot \boldsymbol{e}^{R_i S_k}} = \frac{\sum_k \exp(\boldsymbol{m}^{R_i S_k} - \boldsymbol{n}^{R_i S_k}) \odot \boldsymbol{e}^{R_i S_k} \odot \boldsymbol{u}^{R_i S_k}}{\sum_k \exp(\boldsymbol{m}^{R_i S_k} - \boldsymbol{n}^{R_i S_k}) \odot \boldsymbol{e}^{R_i S_k}}
\end{aligned}$$

with $\boldsymbol{n}^{R_i} \in \mathbb{R}^{H \times |R_i| \times 1}$ being the elementwise maximum of $\boldsymbol{m}^{R_i S_k} \in \mathbb{R}^{H \times |R_i| \times 1}$ over all $1 \leq k \leq \beta$. The fraction is elementwise divison, and expsum is performed row-wise along the last dimension. This expression is numerically stable because each $\boldsymbol{m}^{R_i S_k} - \boldsymbol{n}^{R_i S_k} \leq 0$ and each $\boldsymbol{e}^{R_i S_k} \leq 1$. Now, each aggregate term in the numerator and denominator summations is known to the CompNode. In essence, the CompNode is performing a weighted average of concatenated AttnNode $\boldsymbol{u}$ results, with the weights also coming from AttnNodes $\boldsymbol{m}, \boldsymbol{e}$ results. To get the final output of attention corresponding to row indices in $R_i$, the CompNode finally performs $O$-projection. Algorithm 9 implements this.

---

**Algorithm 7** CompNode$_i$ Single Layer Pre-Pass

---

**input** $\boldsymbol{h}^{R_i}, \boldsymbol{p}^{R_i}$
**output** $\boldsymbol{q}^{R_i}, \boldsymbol{k}^{R_i}, \boldsymbol{v}^{R_i}$
 1: $\boldsymbol{q}^{R_i} \leftarrow$ q_proj($\boldsymbol{h}^{R_i}$)
 2: $\boldsymbol{k}^{R_i} \leftarrow$ k_proj($\boldsymbol{h}^{R_i}$)
 3: $\boldsymbol{v}^{R_i} \leftarrow$ v_proj($\boldsymbol{h}^{R_i}$)
 4: $\boldsymbol{q}^{R_i} \leftarrow$ rotary_pos_emb($\boldsymbol{q}^{R_i}, \boldsymbol{p}^{R_i}$)
 5: $\boldsymbol{k}^{R_i} \leftarrow$ rotary_pos_emb($\boldsymbol{k}^{R_i}, \boldsymbol{p}^{R_i}$)
 6: **return** $\boldsymbol{q}^{R_i}, \boldsymbol{k}^{R_i}, \boldsymbol{v}^{R_i}$

---

---

**Algorithm 8** AttnNode$_{jk}$ Single Layer Attention-Pass

---

**input** $\boldsymbol{q}^{S_j}, \boldsymbol{k}^{S_k}, \boldsymbol{v}^{S_k}, \boldsymbol{s}^{S_j S_k}$
**output** $\boldsymbol{m}^{S_j S_k}, \boldsymbol{e}^{S_j S_k}, \boldsymbol{u}^{S_j S_k}$
 1: $\boldsymbol{k}^{S_k} \leftarrow$ repeat_kv($\boldsymbol{k}^{S_k}$)
 2: $\boldsymbol{v}^{S_k} \leftarrow$ repeat_kv($\boldsymbol{v}^{S_k}$)
 3: $\boldsymbol{a}^{S_j S_k} \leftarrow \boldsymbol{q}^{S_j}(\boldsymbol{k}^{S_k})^T + \boldsymbol{s}^{S_j S_k}$
 4: $\boldsymbol{m}^{S_j S_k} \leftarrow$ row_max($\boldsymbol{a}^{S_j S_k}$)
 5: $\boldsymbol{a}^{S_j S_k} \leftarrow \exp(\boldsymbol{a}^{S_j S_k} - \boldsymbol{m}^{S_j S_k})$
 6: $\boldsymbol{e}^{S_j S_k} \leftarrow$ row_sum($\boldsymbol{a}^{S_j S_k}$)
 7: $\boldsymbol{a}^{S_j S_k} \leftarrow \boldsymbol{a}^{S_j S_k} / \boldsymbol{e}^{S_j S_k}$
 8: $\boldsymbol{u}^{S_j S_k} \leftarrow \boldsymbol{a}^{S_j S_k} \boldsymbol{v}^{S_k}$
 9: **return** $\boldsymbol{m}^{S_j S_k}, \boldsymbol{e}^{S_j S_k}, \boldsymbol{u}^{S_j S_k}$

---

---

**Algorithm 9** CompNode$_i$ Single Layer Post-Pass

---

**input** $\boldsymbol{m}^{R_i S_k}, \boldsymbol{e}^{R_i S_k}, \boldsymbol{u}^{R_i S_k}$ for $1 \le k \le \beta$
**output** $\boldsymbol{o}^{R_i}$
 1: Initialize $\boldsymbol{o}^{R_i}$ with zeroes and shape like $\boldsymbol{u}^{R_i S_1}$
 2: Initialize $\boldsymbol{w}^{R_i}$ with zeroes and shape like $\boldsymbol{e}^{R_i S_1}$
 3: $\boldsymbol{n}^{R_i} \leftarrow$ elementwise_max$\{\boldsymbol{m}^{R_i S_k}\}_{k=1}^{\beta}$
 4: **for** $k = 1$ to $\beta$ **do**
 5: $\quad \boldsymbol{w}^{R_i} \leftarrow \boldsymbol{w}^{R_i} + \exp(\boldsymbol{m}^{R_i S_k} - \boldsymbol{n}^{R_i S_k}) \odot \boldsymbol{e}^{R_i S_k}$
 6: $\quad \boldsymbol{o}^{R_i} \leftarrow \boldsymbol{o}^{R_i} + \exp(\boldsymbol{m}^{R_i S_k} - \boldsymbol{n}^{R_i S_k}) \odot \boldsymbol{e}^{R_i S_k} \odot \boldsymbol{u}^{R_i S_k}$
 7: **end for**
 8: $\boldsymbol{o}^{R_i} \leftarrow \boldsymbol{o}^{R_i} / \boldsymbol{w}^{R_i}$
 9: $\boldsymbol{o}^{R_i} \leftarrow$ o_proj($\boldsymbol{o}^{R_i}$)
10: **return** $\boldsymbol{o}^{R_i}$

---

## H. Integrating Cascade with SMPC

In Section 7.2.1, we concluded that Cascade should only be used when it is safe to reveal some number of tokens, but we mentioned that integration with SMPC could alleviate this requirement. Here, we describe how Cascade can be fused with a number of SMPC schemes to improve token security at the expense of computational and communication cost. The idea is to form an $L$-**layered Cascade-SMPC split**, where a general SMPC scheme is executed on the first $L$ LLM layers and Cascade is carried out thereafter.

We begin with the SMPC stage for layers $\leq L$. Denote the layer 0 input embeddings as $\boldsymbol{x}$, and the layer $L$ hidden states as $\boldsymbol{h}$. For the first $L$ layers, we execute any SMPC protocol $\Phi$ based on *additive secret sharing*. Suppose $\Phi$ involves $t$ nodes $\mathcal{N}_1, \ldots, \mathcal{N}_t$. Then $\Phi$ begins by additively splitting $\boldsymbol{x} = \sum_{s=1}^{t} \boldsymbol{x}_s$, with each share $\boldsymbol{x}_s$ given to $\mathcal{N}_s$. By decomposing LLM layers into SMPC-friendly functions or approximations, $\Phi$ allows each node $\mathcal{N}_s$ to compute some additive share $\boldsymbol{h}_s$ of the layer $L$ hidden states, such that $\boldsymbol{h} = \sum_{s=1}^{t} \boldsymbol{h}_s$. A key ingredient of SMPC schemes based on additive secret sharing is *computational indistinguishability*: no node gains any information about $\boldsymbol{x}$ during the execution of $\Phi$.

Once each $\mathcal{N}_s$ gets $\boldsymbol{h}_s$, we aim to execute Cascade for layers $> L$ in the LLM. We initialize a Cascade sharding scheme $\{R_i\}, \{S_j\}$ and nodes $\{\text{CompNode}_i\}, \{\text{AttnNode}_{jk}\}$, which are some *superset* of the SMPC nodes[6]. Then, for each $i$, each $\mathcal{N}_s$ sends the slice $\boldsymbol{h}_s^{R_i} = \boldsymbol{h}_s[R_i, :] \in \mathbb{R}^{|R_i| \times d}$ to $\text{CompNode}_i$. Finally, $\text{CompNode}_i$ adds its received sliced shares to get $\sum_{s=1}^{t} \boldsymbol{h}_s^{R_i} = \boldsymbol{h}^{R_i}$. Since each $\text{CompNode}_i$ has $R_i$-sharded layer $L$ hidden states, we can execute Cascade for all future layers of the LLM with the Cascade nodes.

In this process, by computational indistiguishability, SMPC nodes gain no information about the input, and thus have no direct token access. Likewise, computational indistiguishability ensures that sliced shares $\boldsymbol{h}_s^{R_i}$ give $\text{CompNode}_i$ no additional information about the input. That is, Cascade nodes no longer have *direct* access to tokens: information leakage only comes from the usual exposed shards at layers $> L$. Thus, even though the execution of $\Phi$ on the first $L$ layers may be more expensive than Cascade, token security is improved. To determine the optimal choice of $L$, further analysis should be conducted on how information leakage from *all* exposed Cascade shards varies across layers.

## I. Breaking Closed-Weight Permutation-Based Privacy-Preserving Schemes

We now describe how our attack in Section 5 can be modified to the closed-weight setting. We break our analysis into three sections: describing the protocols in the closed-weight setting, explaining how our attack applies in these settings if public embedding layer (lookup table) access is assumed, and showing how to relax the lookup table assumption.

### I.1. Closed-Weight Scheme Descriptions

We first give more detail on the original closed-weight implementations of the STIP and Centaur schemes from Section 5.1.

**STIP** In STIP, there are three parties: the model developer $P_1$, the model server $P_2$ (who carries out inference), and the user $P_3$. The goal of STIP is to have $P_2$ carry out inference on $P_3$'s input, protect $P_1$'s private model weights $\Theta$ from $P_2$ and $P_3$, and protect $P_3$'s private input data from $P_1$ and $P_2$. This is done with random permutation in the hidden dimension. At initialization, $P_1$ sends random $d \times d$ permutation matrices $\pi, \pi_c$ to the user $P_3$, where $d$ is the token embedding dimension. They also randomly permute each weight matrix or vector in the row and/or column dimensions, to obtain the altered model weights $\Theta'$; these are given to the model server $P_2$, who cannot recover $\Theta$ from them. Then during inference, instead of sending their private input data $X \in \mathbb{R}^{N \times d}$, the user encrypts it with permutation $\pi$, i.e. they send $X\pi$. Next, a standard transformer forward pass is carried out, but with the weights $\Theta$ (unknown to $P_2$) replaced by permuted weights $\Theta'$. Finally, the results are sent to the user, who applies permutation $\pi_c$ to obtain the output of the inference. The STIP authors show through orthogonality of permutation matrices that the final output obtained is the same output as vanilla inference.

**Centaur** Centaur follows the three-party threat model of STIP, and attempts to reconcile two problems. On the model weight privacy side, they aim to prevent exposure of the lookup table to the user. On the user privacy side, they wish to avoid exposing certain unpermuted intermediate results. For example, the authors observe that during the computation of attention, the calculation of $QK^T$ at each layer in STIP is insecure due to the $Q$ and $K$ permutations canceling. Therefore the authors apply the cryptographically-based technique of *additive secret sharing* between the developer $P_1$ and server $P_2$ at most

---

[6]That is, unless there are more SMPC nodes than Cascade nodes. But this is usually never the case in practice, because most SMPC schemes based on additive secret sharing involve 2 or 3 parties.

stages of self-attention, only requiring reconstruction of additive shares (by the developer) during nonlinearities. Although this resolves the previous two concerns, it is still the case that permutations of true layer $l$ hidden states are exposed to the model developer at nonlinearities.

## I.2. Public Embedding Layer

We now describe the implications of the efficacy of our family of attacks for the above schemes, as well as PermLLM, assuming the embedding layer of the LLM is publicly known.

**PermLLM**     Recall that PermLLM reveals the permuted hidden states at the non-linearities to the parties performing inference; and that the hiddens at the softmax and layer-norm non-linearities, in particular, undergo factorized-2D permutation as they are row-wise operations. Therefore, any party that receives the hidden states at these non-linearities, at any layer, can directly apply the attack described in Section 5. We note that PermLLM claims to avoid such permutation security pitfalls by ensuring that the *user* is one of the parties performing inference, and the only one with access to permuted hidden states. However, such user involvement, while protecting against our attack, makes PermLLM entirely impractical for larger models: the user must still do the bulk of computational cost in matrix multiplications during additive secret sharing.

**STIP**     Recall that in STIP, party $P_2$ carries out inference using a model with permuted weights $\Theta'$, on a permutation of the input, $X\pi$, in the hidden dimension. Apart from an additional detail regarding access to the embedding layer, which we expand on below in Appendix I.3, this is analogous to the hidden-dimension permutation setting. A forward pass from the altered transformer model with weights $\Theta'$ up to layer $l$ will allow $P_2$ to recover hidden-dimension-permuted layer $l$ hidden states, and apply the attack from Section 5 to recover the input.

**Centaur**     Centaur operates similarly to STIP from the perspective of our attack; at the non-linearities, hidden-dimension-permuted hidden states are revealed to the parties performing inference; and party $P_2$ has access to the permuted weights $\Theta'$. Therefore, the attack of Section 5 can also be used on Centaur.

## I.3. Private Embedding Layer

In the above section, we assumed public lookup table access in order to perform forward passes over the vocabulary in the attack. However, in both STIP and Centaur, the party $P_2$, which performs inference, has access to the entire set of permuted model weights $\Theta'$ *except* for the token embedding layer, which is only revealed to the user $P_3$. As such, without direct access to the possible token embeddings, the adversary cannot immediately carry out our attack as described in Section 5. Here, we explain how an adversary can still carry out our attack in the private embedding layer setting, by covering two cases: one where the embedding matrix is tied to the LM (language-modeling) head, and one where it is not.

**Tied Embedding**     In many modern LLM families, the embedding matrix is simply the tranpose of the LM head, whose permutation in row and column dimensions) is known to $P_2$. Therefore, the vocabulary embedding vectors to search over in this case are simply permuted columns of this permuted language-modeling head – and these permutations can be uncovered by matching against the permuted input embedding vectors received from the user at inference.

Explicitly, denoting $W$ as the original $\mathbb{R}^{V \times d}$ embedding matrix, $P_2$ has access to the permuted language-modeling head $\pi_d W^T \pi_V \in \mathbb{R}^{d \times V}$, where $\pi_V, \pi_d$ are $V \times V, d \times d$ permutation matrices. In inference, the user $P_3$ first applies a $d \times d$ permutation $\pi$ on the input embeddings $e_1, \ldots, e_N \in \mathbb{R}^d$; these are rows of $W$. Therefore, $P_2$ sees permuted embedding vectors $e_1\pi, \ldots, e_N\pi \in \mathbb{R}^d$. Now, assuming the uniqueness of *sorted* rows of $W$, each $e_i\pi$ can be obtained by applying the permutation $\pi\pi_d^{-1}$ on exactly one column of $\pi_d W^T \pi_V$. Thus $P_2$ can recover $\pi\pi_d^{-1}$ by looking for a sorted match between the columns of $\pi_d W^T \pi_V$ and each $e_i\pi$. Once obtained, they can compute $\pi\pi_d^{-1}\pi_d W^T \pi_V = \pi W^T \pi_V$, whose columns are precisely all $\pi$-permuted vocabulary embeddings. With these, because the altered transformer forward pass is carried out on $\pi$-permuted embeddings, $P_2$ can carry out our attack on any permuted layer $l$ hidden states it obtains.

To confirm the plausibility of the above, we examined the embedding matrices of Gemma, Llama and Mistral models and found that it is indeed the case that for these modern LLM families, the rows of $W$ are unique even when sorted.

**Non-Tied Embedding**     Even if the language-modeling head is not the transpose of the embedding matrix, $P_2$ can collect the set of sorted input embeddings over the course of many inference requests. After sufficiently many calls, they can then perform our attack by iterating through this collection of embeddings, permuting them to match the initial permuted input embeddings. The only difference in this case is that $P_2$ must wait for more inference requests in order to carry out the attack, rather than being able to perform it immediately.

The final step to decoding by the adversary is then mapping the embeddings back into tokens. Note that this is essentially isomorphic to breaking a simple substitution cipher. Again, by collecting data over many queries and using simple methods such as frequency analysis and positional information, $P_2$ can learn to decode this into the original tokens; substitution ciphers are in general easily broken given sufficient data.

## J. Distance Correlation Does Not Guarantee Permutation Security

We now contextualize statistical arguments on the security of permuted hidden states. In particular, we clarify why they do not anticipate our attack. Both Yuan et al. (2024) and Luo et al. (2024) rely on results from distance correlation theory (Székely et al., 2007) to support their arguments about the security of permuted hidden states, and thus claim their schemes are secure. Citing Zheng et al. (2022), both papers quote the following result:

$$\mathbb{E}_{\pi, W_A \in \mathbb{Z}^{d \times d}} \left[ \text{Discorr}(x, xW_A\pi) \right] \leq \mathbb{E}_{W_B \in \mathbb{Z}^{d \times 1}} \left[ \text{Discorr}(x, xW_B) \right]. \tag{3}$$

Discorr is the distance correlation function and $x \in \mathbb{R}^d$ is the input vector chosen from a data distribution. Here, the expectations are taken over $W_A$ and $W_B$ sampled from standard random normal distributions and $\pi$ sampled uniformly over all $d!$ permutation matrices. This result demonstrates that the expected distance correlation between any vector and the same vector with a random permuted (dimensionality-preserving) linear map applied is less than the expected distance correlation between the vector and the same vector with a 1-dimensional compressing linear map applied. Therefore, the authors claim that permuted LLM hidden states retain less information about the input embeddings than a 1-D projection.

In each the following three subsections, we provide a separate reason for why this result cannot be used to make strong guarantees on the security of their schemes.

### J.1. Reconstruction From Random 1D Projections Is Feasible

The authors assert that reconstructing inputs after a random 1-dimensional linear projection is difficult. However, there is no theoretical reason that this should be the case, especially for such projections of LLM hidden states.

We can make this statement precise as follows. Our attack is able to successfully reverse LLM hidden states with L1-distance matching as demonstrated in Section 4.4. Assuming that two vectors are non-colliding with respect to L1-distance, we can ensure random 1D projections of these two vectors are also non-colliding with high probability.

**Theorem J.1.** *Let $k > 0$. Suppose random weights $\boldsymbol{w} \in \mathbb{R}^d$ are drawn from a $d$-variate spherically symmetric distribution $\mathcal{D}$. Then any $\boldsymbol{x}, \boldsymbol{y} \in \mathbb{R}^d$, we have the absolute difference of $\boldsymbol{w}$-weighted sums of $\boldsymbol{x}$ and $\boldsymbol{y}$ exceeds the L1 distance between $\boldsymbol{x}$ and $\boldsymbol{y}$ by a factor $\geq k$, meaning*

$$\left| \sum_{i=1}^{d} w_i x_i - \sum_{i=1}^{d} w_i y_i \right| \geq k \sum_{i=1}^{d} |x_i - y_i|, \tag{4}$$

*with probability $\geq P_{\boldsymbol{\gamma} \sim \mathcal{D}}(|\gamma_1| \geq k\sqrt{d})$.*

*Proof.* Denote $\boldsymbol{z} = \boldsymbol{x} - \boldsymbol{y}$. Observe that

$$\left| \sum_{i=1}^{d} w_i x_i - \sum_{i=1}^{d} w_i y_i \right| = \left| \sum_{i=1}^{d} w_i(x_i - y_i) \right| = \left| \sum_{i=1}^{d} w_i z_i \right| = |\boldsymbol{w}^T \boldsymbol{z}|.$$

Thus, Equation (4) is equivalent to $|\boldsymbol{w}^T \boldsymbol{z}| \geq k\|\boldsymbol{z}\|_1$. Then, from the standard bound $\|\boldsymbol{z}\|_1 \leq \sqrt{d}\|\boldsymbol{z}\|_2$, which can be proven by an application of Cauchy-Schwarz, we see that Equation (4) holds whenever

$$|\boldsymbol{w}^T \boldsymbol{z}| \geq k\sqrt{d}\|\boldsymbol{z}\|_2. \tag{5}$$

We now aim to compute the probability of the above event. Choose a $d \times d$ orthogonal matrix $Q$ such that $\boldsymbol{z}_q := Q\boldsymbol{z} \in \mathbb{R}^d$ only has a nonzero coordinate $L$ in its first position, i.e. $\boldsymbol{z}_q = (L, 0, \ldots, 0)$. By orthogonality and the fact that $\mathcal{D}$ is spherically symmetric, we see $\boldsymbol{w}_q := Q\boldsymbol{w}$ has distribution $\mathcal{D}$. Furthermore, orthogonal linear transformations are

length-preserving (by L2 norm), so we have $\|\boldsymbol{z}_q\|_2 = \|Q\boldsymbol{z}\|_2 = \|\boldsymbol{z}\|_2 = |L|$. In fact, as $Q^T Q = I$, observe that $\boldsymbol{w}^T \boldsymbol{z} = \boldsymbol{w}^T Q^T Q \boldsymbol{z} = (Q\boldsymbol{w})^T (Q\boldsymbol{z}) = \boldsymbol{w}_q^T \boldsymbol{z}_q$. Hence, Equation (5) becomes

$$|\boldsymbol{w}_q^T \boldsymbol{z}_q| = |L||(\boldsymbol{w}_q)_1| \geq k|L|\sqrt{d}.$$

This is equivalent to saying the first coordinate of $\boldsymbol{w}_q$ has magnitude at least $k\sqrt{d}$. But we showed $\boldsymbol{w}_q$ has distribution $\mathcal{D}$, so the probability that Equation (5) holds is precisely $P_{\boldsymbol{\gamma} \sim \mathcal{D}}(|\gamma_1| \geq k\sqrt{d})$. This is therefore a lower bound on the probability that Equation (4) holds, since we showed Equation (4) holds whenever Equation (5) does. $\qquad\square$

Although the above holds over all spherically symmetric distributions, we can obtain an exact bound above by setting $\mathcal{D}$ to a multivariate Gaussian. That is, for $\boldsymbol{w} = (w_1, \ldots, w_d)$, we i.i.d. sample each $w_i \sim \mathcal{N}(0, \sigma^2)$. Then the lower bound in the theorem is $P_{\boldsymbol{\gamma} \sim \mathcal{D}}(|\gamma_1| \geq k\sqrt{d}) = P_{\gamma \sim \mathcal{N}(0,\sigma)}(|\gamma| \geq k\sqrt{d}) = 2 - 2\Phi(k\sqrt{d}/\sigma)$, where $\Phi$ is the normal CDF. With sufficiently large $\sigma$ or small $k$, we can make this lower bound approach $2 - 2\Phi(0) = 1$. For instance, for $d = 4096$ in Llama-3.1-8B-Instruct, if we sample weights with $\sigma = 1$ (as is done by Zheng et al. (2022) in the statement of Equation (3)), setting $k = 1/64$ gives a lower bound of $2 - 2\Phi(1) \approx 32\%$, and setting $k = 1/32$ gives a lower bound of $2 - 2\Phi(2) \approx 5\%$. To increase $k$ (for a stronger guarantee of non-collision of the weighted sums) while maintaining the probability lower bound, one must proportionally increase the standard deviation $\sigma$ of the random weights.

It is therefore plausible that even with access to random 1D linear projections of LLM hidden states, our attack would be successful. Further work should experimentally verify the efficacy of our attack with randomly-weighted sums, in the presence of non-determinism and other practical implementation considerations.

## J.2. Distance Correlation Misaligns With Reconstructibility

To measure privacy leakage, Zheng et al. (2022) use expected distance correlation. They justify their choice by noting distance correlation is a well-known statistical metric, which represents structural similarity between datasets and is straightforward to estimate. However, as we now show, distance correlation is not a universal measure of how reversible one random variable is from another. To demonstrate this shortcoming, we introduce the notion of '$\delta$-reconstructibility', which captures the ability to recover one variable from another variable up to a given absolute threshold. We define it as follows.

**Definition J.2.** Let $X, Y$ be random variables. We say that $(X, Y)$ is $\delta$-**reconstructible** if there exists a function $f(Y)$ such that $|X - f(Y)| \leq \delta$ almost always.

This notion of $\delta$-reconstructibility is directly tied to privacy in our setting, as the non-determinism described in Section 4 forces us to choose the candidate token within a given absolute threshold. We now show by construction that $\delta$-reconstructibility does not perfectly align with distance correlation: there are $\delta$-reconstructible pairs with a lower distance correlation than non-$\delta$-reconstructible pairs.

**Theorem J.3.** *For any $\delta > 0$, there exist random variables $W, X, Y, Z$ such that $Discorr(W, X) > Discorr(Y, Z)$, the pair $(W, X)$ is not $\delta$-reconstructible, and the pair $(Y, Z)$ is $\delta$-reconstructible.*

*Proof.* Define independent random variables $W, \varepsilon \sim \mathcal{N}(0, 1)$. Let $Y$ come from an arbitrary symmetric distribution about zero with support $[-\delta, \delta]$, and construct

$$X = \rho W + \sqrt{1 - \rho^2}\varepsilon, \quad Z = |Y|$$

where $1 > \rho > 0.945$. Using standard properties of normal random variables, one can see $X \sim \mathcal{N}(0, 1)$, and the correlation between $X$ and $W$ is $\rho$. Thus, by Theorem 7 in Székely et al. (2007), which lower bounds distance correlation of standard normals in terms of (Pearson) correlation, we have $\text{DisCorr}(W, X) > 0.89\rho > 0.841$. Furthermore, by Theorem 1 in Edelmann et al. (2021), which upper bounds the distance correlation of a symmetric random variable and its absolute value, we have $\text{DisCorr}(Y, Z) \leq 2^{-1/4} < 0.841$. Therefore, we have $\text{DisCorr}(W, X) > \text{DisCorr}(Y, Z)$.

Now, we claim that $(W, X)$ is not $\delta$-reconstructible. To see this, note $(W, \epsilon) \sim \mathcal{N}(0, I)$ by independence, so the linear transformation $(W, \epsilon) \mapsto (W, X)$ can be seen to induce the joint distribution

$$(W, X) \sim \mathcal{N}\left(0, \Sigma = \begin{pmatrix} 1 & \rho \\ \rho & 1 \end{pmatrix}\right).$$

From the standard conditional Gaussian formula, one obtains $W|(X = x) \sim \mathcal{N}(\rho x, \sqrt{1 - \rho^2})$. Thus, for any estimator $f(X)$ of $Y$, we have for each $x$ that

$$P\left(|W - f(X)| \leq \delta | X = x\right) \leq P\left(|W - \rho x| \leq \delta | X = x\right) = 2\Phi\left(\frac{\delta}{\sqrt{1 - \rho^2}}\right) - 1 = c < 1$$

where $c$ is a constant dependent on $\rho, \delta$, and $\Phi$ is the normal CDF. Here, the first inequality holds as $W|(X = x)$ is a normal distribution: this means $P\left(|W - f(X)| \leq \delta | X = x\right)$, the integral of the corresponding normal PDF over $(f(x) - \delta, f(x) + \delta)$, is upper bounded by its integral over the same-size mean-centered interval $(\rho x - \delta, \rho x + \delta)$, which is precisely $P\left(|W - \rho x| \leq \delta | X = x\right)$. This upper bound follows from the fact that an integral of a zero-centered normal (or generally any unimodal symmetric distribution) over a fixed-size interval is maximal when that interval is zero-centered, which is standard: see the first sentence in Anderson (1955), for example. Finally, taking the expectation of the above bound over $X$ and applying the law of total expectation, we get $P(|W - f(X)| \leq \delta) \leq c < 1$. Since $f(X)$ was chosen arbitrarily, this shows $(W, X)$ is not $\delta$-reconstructible[7], as required for the claim.

However, $(Y, Z)$ is certainly $\delta$-reconstructible. Because $|Y| \leq \delta$ almost always, we see $f(Z) = 0$ always estimates $Y$ within a $\delta$-threshold. Hence, we have our desired counterexample. □

Additionally, we observe that Equation (3) involves an expectation of distance correlation over random linear maps and permutations. Therefore, it is possible that there are particular linear weights and permutations where the distance correlation with a randomly permuted linear projection is smaller than the distance correlation with a random 1D linear projection. So, Equation (3) cannot be applied to make universal claims about reconstructibility across different models and permutations.

### J.3. Transformers Have Token Interdependence

Even taking Equation (3) at face value, it is still questionable how it proves security for *transformer models*. Linear projections are only one component of these architectures: a formal security guarantee should incorporate the other modules in a transformer, especially self-attention, in which tokens are not processed independently. In particular, this means a valid result should be proved over a distribution over full $N \times d$ inputs, rather than a distribution of $1 \times d$ embeddings as in Equation (3). In fact, the unidirectional nature of decoder-only LLMs through self-attention is a key property that enables the vocabulary-matching attack to succeed (Section 4.2.1). Thus, the distance correlation result, which ignores this dependence, fails to anticipate such an attack.

## K. BLEU Scores for CompNode Hidden Reversal

In Section 7.2.2, we computed ROUGE-L scores to evaluate the success of learning-based reversal attacks on shards $h^{R_i}$ received by CompNodes at layer 1. Here, we further assess reconstruction quality at layer 1 using BLEU scores, shown in Table 14. We find a similar trend as in Table 5, observing that security improves as $c$ or $\alpha$ increases. Here, for $c \geq 4, \alpha = 8$, we have a BLEU score between $0.1$ and $0.2$, indicating only marginal reconstructionability; and for $c \geq 4, \alpha = 12$, the BLEU score lies below $0.1$, indicating nearly no reconstructionability.

*Table 14.* BLEU scores of text reconstruction from layer 1 hiddens of Gemma-2-2B-IT for different values of $c, \alpha$ under $(c, \delta)$-sharding.

|         | $\alpha = 4$ | $\alpha = 8$ | $\alpha = 12$ |
|---------|--------------|--------------|---------------|
| $c = 1$ | 0.537        | 0.229        | 0.130         |
| $c = 4$ | 0.352        | 0.133        | 0.086         |
| $c = 8$ | 0.246        | 0.123        | 0.083         |

---

[7]In fact, it shows something stronger: the optimal estimator's probability of reconstructing $W$ up to an absolute error of $\delta$ is upper bounded by $c$. As $\delta \to 0$, the value of $c$ actually approaches $2\Phi(0) - 1 = 0$.

## L. CompNode Hidden Reversal Analysis on Layers & Llama

In Section 7.2.2, we described experiments performed on the hidden states of Gemma-2-2B-IT, where a bidirectional-attention model was trained to reverse the sharded hidden states into the original text prompt. In Table 5, we showed that the hiddens are largely secure to this attack for a suitable choice of $c$ and $\alpha$.

In this section, we first analyze if this is also true for Llama 3.1 8B-Instruct. We run a similar experimental setup as described in Section 7.2.2, except we use Llama hidden representations, and we also use it as the reversal model; this also therefore tests if increasing the reversal model's capacity is a suitable method for improving sharded reconstruction. Due to the computational constraints of training with a larger model, we examine this only for $c = 8, \alpha = 8$ and $c = 8, \alpha = 12$. The reconstruction ROUGE-L scores are 0.1718 and 0.1443 respectively, significantly lower than those obtained with the same parameters for Gemma. We leave to future work the interesting question of whether this implies that Llama representations are inherently more resistant to decoding than Gemma representations.

Next, we analyze whether our results hold irrespective of the layer of the model used. We run additional experiments on the hiddens of layers 11 and 21 of Gemma-2-2B-IT. Our results are shown in Table 15. We see that there is no substantial difference in ROUGE-L score as the layer changes.

*Table 15.* ROUGE-L scores of text reconstruction from the hiddens of various layers of Gemma-2-2B-IT for different $(c, \delta)$-sharding setups. We see that the reconstruction quality is similar across layers.

| Layer | $c = \alpha = 4$ | $c = \alpha = 8$ |
|:-----:|:----------------:|:----------------:|
| 1     | 0.4268           | 0.2218           |
| 11    | 0.4627           | 0.2467           |
| 21    | 0.4021           | 0.2158           |

## M. AttnNode Security

First, we justify our point from Section 7.1 that $S$ and $T$ sharding can be made symmetric at no loss of AttnNode security.

$S$-$T$ **Symmetrization**     Symmetry relaxation arises from a pairwise coverage requirement of $S$ and $T$ sharding: recall that $\{S_j\}_{j=1}^{\beta}$ and $\{T_k\}_{k=1}^{\gamma}$ cover $[N]$, so all index pairs $(x, y) \in [N] \times [N]$ lie in some $S_j \times T_k$. Because $S$ and $T$ sharding correspond to query and key/value sharding, this means for all such index pairs $(x, y)$, there is some $\text{AttnNode}_{jk}$ that holds both the $x$th query row and the $y$th key row. Observe that a node holding these rows has – in the worst case – the same information as if they hold the $y$th query and $x$th key rows. This is because the query and key matrices are linear projections, which are injective and reversible in the worst-case, of the same hidden states. Therefore, we justify our simplification in Section 7.1 that $T$ sharding may be set equal to $S$ sharding without security loss.

Now, we introduce a technique that allows us to improve AttnNode security. Since we have decided to set $S$ and $T$ sharding equal, we would like a secure way to select $S$ sharding, given $R$ sharding. One option is to set $S$ and $R$ sharding equal, and have each $\text{AttnNode}_{jk}$ receive the union of query shards from $\text{CompNode}_j$ and key/value shards from $\text{CompNode}_k$. However, this results in each AttnNode having access to twice as many query/key/value rows as each CompNode. To reduce such leakage to AttnNodes and thus improve security, we instead propose a further $m$-split of AttnNodes as follows.

$m$-**Splitting of AttnNodes**     To form $S$ sharding from $R$ sharding, we can let $\beta = m\alpha$, and partition each $R_i$ into $m$ shards $R_{i,1}, \ldots, R_{i,m}$. Then the $S$ shards are $S_{m(i-1)+x} = R_{i,x}$ for all $1 \le i \le \alpha$ and $1 \le x \le m$. This ensures that pairwise coverage is maintained, but reduces the number of tokens (technically $Q, K, V$-projections of hidden state rows, but these are reversible to tokens in the worst case at layer 0) that each AttnNode has access to by a factor of $m$. Using this split increases the value of $\beta^2$, the total number of AttnNodes, by a factor of $m^2$.

$m$-**Splitting for** $(c, \delta)$ **Sharding**     We gave a general construction: there remains a degree of freedom in deciding the exact choice of subdividing $R_i$ into the subsets $R_{i,x}$. Under the assumption that $R_i$ follows a $(c, \delta)$-sharding scheme, we propose that $R_{i,x}$ contains the elements of sorted $R_i$ at indices $x, x + \delta, \ldots, x + (t - 1)\delta$, where $t = \frac{N}{c\alpha}$ and the split factor is $m = c$.

For example, suppose that $\alpha = 3$, $c = 2$, $\delta = 6$, and $N = 18$. With a split of $m = 2$, we have $\beta = m\alpha = 6$. Then,

$$R_1 = \{1, 2, 7, 8, 13, 14\} \qquad R_2 = \{3, 4, 9, 10, 15, 16\} \qquad R_3 = \{5, 6, 11, 12, 17, 18\}$$
$$R_{1,1} = \{1, 7, 13\} \qquad R_{2,1} = \{3, 9, 15\} \qquad R_{3,1} = \{5, 11, 17\}$$
$$R_{1,2} = \{2, 8, 14\} \qquad R_{2,2} = \{4, 10, 16\} \qquad R_{3,2} = \{6, 12, 18\}$$
$$S_{1111} = \{1, 7, 13\} \qquad S_{1121} = \{1, 3, 7, 9, 13, 15\} \qquad S_{1131} = \{1, 5, 7, 11, 13, 17\}$$
$$S_{1112} = \{1, 2, 7, 8, 13, 14\} \qquad S_{1122} = \{1, 4, 7, 10, 13, 16\} \qquad S_{1132} = \{1, 6, 7, 12, 13, 18\}$$
$$S_{2121} = \{3, 9, 15\} \qquad S_{2131} = \{3, 5, 9, 11, 15, 17\} \qquad S_{2112} = \{2, 3, 8, 9, 14, 15\}$$
$$S_{2122} = \{3, 4, 9, 10, 15, 16\} \qquad S_{2132} = \{3, 6, 9, 12, 15, 18\} \qquad S_{3131} = \{5, 11, 17\}$$
$$S_{3112} = \{2, 5, 8, 11, 14, 17\} \qquad S_{3122} = \{4, 5, 10, 11, 16, 17\} \qquad S_{3132} = \{5, 6, 11, 12, 17, 18\}$$
$$S_{1212} = \{2, 8, 14\} \qquad S_{1222} = \{2, 4, 8, 10, 14, 16\} \qquad S_{1232} = \{2, 6, 8, 12, 14, 18\}$$
$$S_{2222} = \{4, 10, 16\} \qquad S_{2232} = \{4, 6, 10, 12, 16, 18\} \qquad S_{3232} = \{6, 12, 18\}$$

where we denote $S_{ixjy} = R_{i,x} \cup R_{j,y}$, which is the set of tokens that AttnNode$_{a,b}$ has access to for $a = m(i-1) + x$, $b = m(j-1) + y$. In other words, $R_i$ above are the sets of token indices that the CompNodes receive, and the $S_{ixjy}$ are the sets of token indices that the AttnNodes receive. Note that some $S_{ixjy}$ entries are not included above (there are 21 listed, but $\beta^2 = 36$ AttnNodes) because they exactly match a listed entry by $S_{ixjy} = S_{jyix}$.

**Vocab-Matching Attack** Sharding $S$ in this way prevents vocab-matching, like in Section 7.2.2. Indeed, as each $R_{i,x}$ has elements that are separated by $\delta$, and each $S_{ixjy}$ combines elements from two different $R_{i,x}$'s, then there cannot be 3 consecutive elements in $S_{ixjy}$ if $\delta > 2$. Furthermore, the largest number of missing tokens between two elements of $S_{ixjy}$ (i.e. the largest 'gap') is lower bounded by $\frac{\delta}{2}$. Therefore, for sufficiently large $\delta$, the vocab-matching attack is infeasible.

**Learning-Based Attacks** To test security against learning-based attacks, we conducted experiments with the above scheme for $m = \{2, 3, 4\}$, with $c = 8$ and $\alpha = 8$, and the same dataset and setup as in Section 7.2.2. Due to computational constraints, we focus our experiments on Gemma-2-2B-IT on layer 1; we expect similar trends for Llama-3.1-8B-Instruct and other layers. We train a single model for all shard possibilities that arise from $S_{ixjy}$. Experiments are conducted with the same dataset and model setup as described . Our results are shown in Table 16. We see that although $m = 2$ results in a relatively higher ROUGE-L than that for the CompNodes in Table 5, the score for $m = 4$ is very similar; therefore, we recommend using $m \geq 4$ for security.

*Table 16.* ROUGE-L scores for different values of splitting parameter $m$ on layer 1 of Gemma-2-2B-IT with $c = 8, \alpha = 8$. We see that the score for $m = 4$ is similar to that of CompNodes in Table 5 for the same $c$ and $\alpha$.

| $m$ | ROUGE-L |
| --- | --- |
| 2 | 0.3057 |
| 3 | 0.2643 |
| 4 | 0.2376 |

# N. Computational Overhead Analysis

In Section 7.3.1, we claimed that Cascade has little overhead in computational costs compared to vanilla inference. We justify this statement in the analysis below, by comparing CompNode and AttnNode steps against the vanilla forward pass. For simplicity of analysis, we assume symmetry of $S$ and $T$ sharding, as justified in Appendix M.

Indeed, most operations performed by CompNodes will treat the (row) token dimension as the batch dimension. In the pre-pass, these are normalization and $Q, K, V$-projection; and in the post-pass, these are attention value compilation (most of Algorithm 9), $O$-projection, residual connection, and the MLP block. Except for attention value compilation, these steps all occur in the vanilla pass, so the CompNodes combined will perform the exact same operations as in vanilla inference: there are no extra computations performed.

The only extra operations thus come from **(a)** attention value compilation (linear weighting of partial attention outputs) by CompNodes in the post-pass, and **(b)** AttnNode floating point computations which do not appear in the vanilla pass, i.e.

expsums of shards of attention score rows. This is because all other steps of the Cascade self-attention either treat the tokens as batch elements, or involve splitting up matrix multiplication into multiplication of sharded matrices; and the latter is blocked matrix multiplication, which does not inherently change the operations performed.

Now, **(a)** only involves $\sim H|R_i|d$ operations for each $\text{CompNode}_i$, since it involves a few steps of elementwise summation and multiplication of $H \times |R_i| \times d$ matrices (after broadcasting). Summing this over all $1 \le i \le \alpha$ gives $\sim \sum_i H|R_i|d = HNd$ extra operations. Also, **(b)** only involves $\sim H|S_j||S_k|$ operations for each $\text{AttnNode}_{jk}$ because expsum is done over rows of an $H \times |S_j| \times |S_k|$ shard of attention scores. Summing over all $1 \le j, k \le \beta$, we see this requires $\sim \sum_{j,k} H|S_j||S_k| = HN^2$ extra operations in total. This means the total AttnNode computation overhead is $\sim HN(d+N)$ operations.

Importantly, this is cheaper than most computation-heavy steps in standard inference. Compared to the $\sim HN^2d$ operations from multiplication of $H \times N \times N$ attention scores with $H \times N \times d$ values, this overhead requires $\sim \frac{1}{N} + \frac{1}{d}$ times as many operations. Since $d$ is often in the hundreds, we can ensure for large $N$, say $N \ge 256$, that this ratio is quite small. Furthermore, if $N$ is not large, then the overhead is still limited compared to the $\sim HNd_{emb}d$ operations from $Q, K, V$-projection, since it requires $\sim \frac{1}{d_{emb}} + \frac{N}{d_{emb}}$ times as many operations and $d_{emb}$ is in the hundreds or thousands. Essentially, the choice of sharding does not significantly affect the total computational overhead, and this overhead is quite modest compared to the computations performed in a vanilla forward pass.

## O. Communication Analysis

In Section 7.3.1, we gave the total communication byte and time overheads for performing a inference on a single layer of an LLM with Cascade. Here, we provide a full justification of these equations. Like in Appendix N, we assume symmetry of $S$ and $T$ sharding, so the superscript $T$ in sharded notation is replaced with $S$.

Recall that in each layer, there are two communication stages: **(A)** the CompNodes send sharded query, key, value matrices to the AttnNodes between pre-pass and attention-pass, and **(B)** the AttnNodes send sharded attention outputs and expsums to the CompNodes between attention-pass and post-pass. We operate under the assumption that all CompNodes synchronize before **(A)** and all AttnNodes synchronize before **(B)**, so that we can derive an exact expression for communication cost; this makes our communication cost derivation a worst-case analysis. See Appendix Q for optimizations that can be made if this assumption is relaxed.

For single-layer inference, in stage **(A)**, $\text{CompNode}_i$ must send each of the $|R_i|$ rows of the $H \times |R_i| \times d$ query matrix $\boldsymbol{q}^{R_i}$ to some AttnNodes. In particular, for a row index $r \in R_i$, it sends the row $\boldsymbol{q}[:, r, :]$ of $\boldsymbol{q}^{R_i}$ to all $\text{AttnNodes}_{j_r k}$ with $1 \le k \le \beta$, where $j_r$ is the unique index satisfying $r \in S_{j_r}$. That is, $\text{CompNode}_i$ sends each of its $|R_i|$ rows to exactly $\beta$ AttnNodes. Since each row contains $Hd$ elements, then $\text{CompNode}_i$ must send out $\beta|R_i|Hd$ elements from sharded query states. A similar analysis shows $\text{CompNode}_i$ sends out $2\beta|R_i|H_{KV}d$ elements from sharded key and value states, so it sends out a total of $\beta|R_i|d(H + 2H_{KV})$ elements. Summing this over all $i$ and noting $\sum_{i=1}^{\alpha}|R_i| = N$ gives the total bytes communicated in **(A)**:

$$\text{CommBytes}_A = \beta Fd(H + 2H_{KV}) \cdot N.$$

Assuming perfect parallel transport and uniform bandwidth $B$ across nodes, i.e. all communication overhead comes from CompNode with the most elements to send (plus latency $\tau$), the communication time in stage **(A)** is

$$\text{CommTime}_A = \tau + \frac{\beta Fd(H + 2H_{KV})}{B} \cdot \max_i |R_i|.$$

Next, in stage **(B)**, each $\text{AttnNode}_{jk}$ must send each $\text{CompNode}_i$ some rows of its partial post-value attention outputs $\boldsymbol{u}^{S_j S_k}$, partial attention score row maximums $\boldsymbol{m}^{S_j S_k}$, and partial attention score row subtract-max expsums $\boldsymbol{e}^{S_j S_k}$. These matrices are of shapes $H \times |S_j| \times d, H \times |S_j| \times 1, H \times |S_j| \times 1$, respectively, and $\text{CompNode}_i$ receives $|R_i \cap S_j|$ out of the $|S_j|$ rows from each. This means the total number of elements that $\text{AttnNode}_{jk}$ sends to all CompNodes is

$$(d+2)H \cdot \sum_{i=1}^{\alpha} |R_i \cap S_j| = (d+2)H \cdot |S_j|.$$

Since $\sum_{j,k=1}^{\beta} |S_j| = \beta \sum_{j=1}^{\beta} |S_j| = \beta N$, this means the total number of bytes sent by all $\beta^2$ AttnNodes is

$$\text{CommBytes}_B = \beta F(d+2)H \cdot N.$$

And, again under the parallel transport and uniform bandwidth assumption, the communication time in **(B)** is

$$\text{CommTime}_B = \tau + \frac{F(d+2)H}{B} \cdot \max_j |S_j|.$$

Combining these costs, we obtain the following total communication byte and time overheads for a single layer:

$$\text{CommBytes} = \beta F(2dH + 2dH_{KV} + 2H) \cdot N,$$
$$\text{CommTime} = 2\tau + \frac{\beta F d(H + 2H_{KV})}{B} \cdot \max_i |R_i| + \frac{F(d+2)H}{B} \cdot \max_j |S_j|.$$

Finally, we compute the number of communication rounds per layer. Stage **(A)** has each of the $\alpha$ CompNodes send results to at most $\beta^2$ AttnNodes, which is at most $\alpha\beta^2$ rounds. Stage **(B)** has each of the $\beta^2$ AttnNodes send results to at most $\alpha$ CompNodes, which is at most $\alpha\beta^2$ rounds. In total, the rounds per layer are bounded above by $2\alpha\beta^2$. This can be quite large, but we can guarantee a tighter upper bound if our scheme involves $(c, \delta)$-sharding for CompNodes with $m$-splitting of AttnNodes (as in Appendix M). Here, $\beta = m\alpha$ since each of the $\alpha$ shards in $\{R_i\}_{i=1}^{\alpha}$ is split into $m$ pieces to form $\{S_j\}_{j=1}^{\beta}$. Each CompNode sends results to $m\beta$ AttnNodes, and each AttnNode sends results to 1 CompNode, so there are $m\alpha\beta + \beta^2 = 2\beta^2$ rounds. Essentially, the number of rounds scales linearly with the number of AttnNodes.

## P. Full CompNode Leakage Analysis

### P.1. Necessity of $(c, \delta)$-Sharding

We have examined information leakage from $h^{R_i}$ in Section 7.2.1 and Section 7.2.2, as well as from $q^{S_j}, k^{S_k}, v^{S_k}$ in Section 7.2.3. For a comprehensive security analysis, all that remains is to consider additional leakage from $m^{R_i S_1}, e^{R_i S_1}, u^{R_i S_1}, \dots, m^{R_i S_\beta}, e^{R_i S_\beta}, u^{R_i S_\beta}$, the information that CompNode$_i$ receives from $\beta$ AttnNodes to begin the post-pass. Our main theorem shows that considering this leakage, a scheme like $(c, \delta)$-sharding is, in some sense, required to maintain security against vocab-matching: without sufficiently large gaps between consecutive clusters of indices, a variant of the attack can be carried out at Layer 0 to reveal additional tokens to a node.

**Theorem P.1.** *Suppose $S$ and $T$ sharding are equal, and the attention mask $\boldsymbol{s}$ is unidirectional. Furthermore, denote the vocab-matching threshold as $\rho$, as defined in Definition 7.2. Then to prevent the vocab-matching attack, for all $i \in [\alpha]$, each gap between clusters of consecutive indices in $R_i$ must have size $\geq \rho$.*

*Proof.* Fix $i$, and denote $R_i = \{r_1, r_2, \dots, r_m\}$ in ascending order. Consider any $k \in [\beta]$. Now, since $\boldsymbol{a}^{R_i S_k} = \boldsymbol{q}^{R_i}(\boldsymbol{k}^{S_k})^T + \boldsymbol{s}^{R_i S_k}$, and $\boldsymbol{s}$ is unidirectional, we see that the $l$th row of $\exp(\boldsymbol{a}^{R_i S_k})\boldsymbol{v}^{S_k}$ is exactly $f(r_l, \{s \in S_k : s < r_l\})$, where we denote

$$f(r, S) := \sum_{s \in S} \exp\left(\boldsymbol{q}[:, r, :](\boldsymbol{k}[:, s, :])^T\right)\boldsymbol{v}[:, s, :].$$

This $l$th row is known to CompNode$_i$ because we can write it in terms of known shards:

$$\exp(\boldsymbol{a}^{R_i S_k})\boldsymbol{v}^{S_k} = (\text{softmax}(\boldsymbol{a}^{R_i S_k})\boldsymbol{v}^{S_k}) \odot \text{expsum}(\boldsymbol{a}^{R_i S_k}) = \boldsymbol{u}^{R_i S_k} \odot \boldsymbol{e}^{R_i S_k} \odot \exp(\boldsymbol{m}^{R_i S_k}).$$

Thus, for each $r_l \in R_i$ and $k \in [\beta]$, we see that CompNode$_i$ knows $f(r_l, \{s \in S_k : s < r_l\})$. At Layer 0, since $\boldsymbol{q}, \boldsymbol{k}, \boldsymbol{v}$ are linear projections of token embeddings, we see that $f(r, S)$ only depends on tokens with indices in the set $\{r\} \cup S$. Since CompNode$_i$ knows all tokens with indices $R_i$ from $\boldsymbol{h}^{R_i}$ at layer 0, then each $f(r_l, \{s \in S_k : s < r_l\})$ only depends on *unknown* tokens at indices $\{s \in S_k : s < r_l\} \setminus R_i$. Thus, $f(r_{l+1}, \{s \in S_k : s < r_{l+1}\})$ depends on the same unknown tokens as $f(r_l, \{s \in S_k : s < r_l\})$, and extra unknown tokens at $\{s \in S_k : r_l < s < r_{l+1}\}$, which we call the $(k, l)$-gap.

If there are $< \rho$ tokens in a $(k, l)$-gap for some $l$, then following the generalized attack in Appendix C, CompNode$_i$ can perform a forward pass over all $V^{<\rho}$ candidate sequences of such unknown tokens in the $(k, l)$-gap and compute candidate values for $f(r_l, \{s \in S_k : s < r_l\})$ and $f(r_{l+1}, \{s \in S_k : s < r_{l+1}\})$, and then select the candidate sequence which allows these quantities to match their known values. This would allow them to recover all tokens in the $(k, l)$-gap. Thus, to prevent this vocab-matching attack variant, we need each $(k, l)$-gap to have size $\geq \rho$ or zero. This forces clusters of consecutive indices in $R_i$ to have size $\geq \rho$ gaps between them. $\square$

## P.2. Sufficiency of $(c, \delta)$-Sharding

Now, we provide an argument that only considering Layer 0 shards, when $R$ and $S$ sharding both follow a $(c, \delta)$-sharding scheme with sufficiently large $c, \delta$, it is intractable for any CompNode to recover the input.

To do this, we will demonstrate a reduction to a variant of the subset sum problem with vectors, which is known to be NP-complete. Our reduction relies on two assumptions **(B1)**, **(B2)**, which we highlight shortly. We begin by fixing $i \in [\alpha]$, so that $R_i = \{ic, ic+1, \ldots, ic+c-1, ic+\delta, ic+\delta+1, \ldots, ic+\delta+c-1, \ldots\}$. As we mentioned earlier, CompNode$_i$ has access to $\boldsymbol{m}^{R_i S_1}, \boldsymbol{e}^{R_i S_1}, \boldsymbol{u}^{R_i S_1}, \ldots, \boldsymbol{m}^{R_i S_\beta}, \boldsymbol{e}^{R_i S_\beta}, \boldsymbol{u}^{R_i S_\beta}$ and $\boldsymbol{h}^{R_i}$ at Layer 0. Knowledge of the latter is equivalent to knowledge of tokens at indices $R_i$, so we consider leakage from the former triples. In fact, at Layer 0, note each triple $\boldsymbol{m}^{R_i S_k}, \boldsymbol{e}^{R_i S_k}, \boldsymbol{u}^{R_i S_k}$ only depends on tokens with indices in $R_i \cup S_k$. Since all tokens in $R_i$ are known to CompNode$_i$, then this triple only depends on *unknown* tokens $S_k \setminus R_i$. Across all $k$, these sets are disjoint. Thus, under the following assumption, we can conclude that the reversal problems from the $\beta$ different triples are *independent*.

**(B1)** For any $k_1 \neq k_2$, CompNode$_i$ has independent priors on tokens with indices in $S_{k_1}$ and tokens with indices in $S_{k_2}$.

Thus, we now only need to consider leakage from *one* triple $\boldsymbol{m}^{R_i S_k}, \boldsymbol{e}^{R_i S_k}, \boldsymbol{u}^{R_i S_k}$. Explicitly, $S_k = \{kc, kc+1, \ldots, kc+c-1, kc+\delta, kc+\delta+1, \ldots, kc+\delta+c-1, \ldots\}$. So, if $k = i$ and $S_k = R_i$, then all these shards are functions only of *known* tokens at indices $R_i$, and no information is leaked to CompNode$_i$. Thus, without of loss of generality, we now assume $k < i$, as $k > i$ is similar. From now on, when we say a shard depends on tokens, we ignore known tokens.

Recall that $\exp(\boldsymbol{a}^{R_i S_k})\boldsymbol{v}^{S_k} = \boldsymbol{u}^{R_i S_k} \odot \boldsymbol{e}^{R_i S_k} \odot \exp(\boldsymbol{m}^{R_i S_k})$, and $\text{expsum}(\boldsymbol{a}^{R_i S_k}) = \boldsymbol{e}^{R_i S_k} \odot \exp(\boldsymbol{m}^{R_i S_k})$ by definition. Thus, this triple reveals the exact same information as the triple $\boldsymbol{m}^{R_i S_k}, \text{expsum}(\boldsymbol{a}^{R_i S_k}), \exp(\boldsymbol{a}^{R_i S_k})\boldsymbol{v}^{S_k}$, which have shapes $(H, |R_i|, 1), (H, |R_i|, 1), (H, |R_i|, d)$. We concatenate the last two to form $\boldsymbol{b}^{R_i S_k}$ of shape $(H, |R_i|, d+1)$, so our original triple is equivalent to $\boldsymbol{m}^{R_i S_k}, \boldsymbol{b}^{R_i S_k}$. Now, note that because $S$ and $R$ are both $(c, \delta)$-sharding and the attention mask is unidirectional, the first $c$ out of $|R_i|$ rows of $\boldsymbol{a}^{R_i S_k}$ will have $c$ elements followed by all $-\infty$, with the nonzero $c$ only depending on tokens at indices $kc, \ldots, kc+c-1$, respectively. The next $c$ rows have $2c$ elements that depend on tokens at indices $kc, \ldots, kc+c-1, kc+\delta, kc+\delta+1, \ldots, kc+\delta+c-1$, followed by all $-\infty$. A similar pattern holds for each next $c$ rows. Thus, the $|R_i|$ rows of $\boldsymbol{b}^{R_i S_k}$ follow a similar token dependence pattern: the first $c$ depend on tokens $kc, \ldots, kc+c-1$, the next $c$ depend on tokens $kc, \ldots, kc+c-1, kc+\delta, kc+\delta+1, \ldots, kc+\delta+c-1$, and so on.

Now, for each position $x \in [N]$, we define a list of $V$ possible concatenations $[k_x, v_x] \in \mathbb{R}^{H \times 2d}$, where $k_x$ denotes the $x$th row across all heads of the layer 0 query states, and likewise for $v_x$. Denote this list as $\mathcal{A}_x$, which is distinct for different values of $x$ due to the effects of positional embeddings; it is indexed in the same order as the vocabulary. For each $r \in R_i$ and position $x \in [N]$ with $x \leq r_i$, we denote $\mathcal{B}_{r,x}$ as the list of $(d+1)$-dimensional vectors whose first entry is $\exp(\boldsymbol{q}[:, r, :]k_x^T)$ and last $d$ entries are $\exp(\boldsymbol{q}[:, r, :]k_x^T)v_x$, over all $[k_x, v_x] \in \mathcal{A}_x$. Note this is also of size $V$, and is indexed in the same order as the vocabulary. It is simple for a node to compute $\mathcal{B}_{r,x}$ for all $r \in R_i$ and $x \leq r_i$, as they know $\boldsymbol{q}[:, r, :]$, and can directly compute and iterate through $\mathcal{A}_x$.

The key point, building on our observation from earlier, is that $(d+1)$-dimensional rows of $\boldsymbol{b}^{R_i S_k}$ are sums of vectors from sets $\mathcal{B}_{r,x}$. In particular, there are $i_1, \ldots, i_c$ such that for all $1 \leq r \leq c$, the $r$th row equals $\mathcal{B}_{r,kc}[i_1] + \ldots + \mathcal{B}_{r,kc+c-1}[i_c]$. Then, there are $i_{c+1}, \ldots, i_{2c}$ such that for all $c+1 \leq r \leq 2c$, the $r$th row equals $\mathcal{B}_{r,kc}[i_1] + \ldots + \mathcal{B}_{r,kc+c-1}[i_c] + \mathcal{B}_{r,kc+\delta}[i_{c+1}] + \ldots + \mathcal{B}_{r,kc+\delta+c-1}[i_{2c}]$. A similar pattern holds for $2c+1 \leq r \leq 3c$, and so on. The final indices $i_1, i_2, \ldots$ correspond to the positions of input tokens in the vocabulary, for all tokens with indices in $S_k$.

Therefore, reversal of the input from $\boldsymbol{b}^{R_i S_k}$ alone is at least as difficult as the following problem: given $c$ different sets, each containing $V$ vectors derived from token and position embeddings, select some vector from each set so that they sum to a given vector. This is a variant of the subset sum problem for vectors, which is computationally intractable for large enough $V, c$. Particularly, since $V$ is in the hundreds of thousands, this is likely already intractable for $c \geq 8$.

Finally, we consider additional leakage from $\boldsymbol{m}^{R_i S_k}$. In the worst case, this reveals at most one of the tokens with indices in $S_k$ for each $k$, as a maximum of a row of $\boldsymbol{a}^{R_i S_k}$ reveals at most one element of the row, with upper bound constraints placed on the other elements. Given the large number of possibilities for these elements, it is unlikely that the upper bound constraints provide much information, which we incorporate in the assumption below. Therefore, leakage from $\boldsymbol{m}^{R_i S_k}$ is at worst equivalent to effectively reducing the parameter $c$ above by one, since we have one less set to select a vector from. This means if $c \geq 8$ was secure for reversal from $\boldsymbol{b}^{R_i S_k}$, we need $c \geq 9$ to ensure security for reversal from $\boldsymbol{b}^{R_i S_k}, \boldsymbol{m}^{R_i S_k}$. Note that the variant of the subset sum problem for reversal from $\boldsymbol{b}^{R_i S_k}$, together with the inequality constraints from $\boldsymbol{m}^{R_i S_k}$ leakage, form a linear program. So, we have implicitly used the following assumption to claim security.

**(B2)** The linear program described above is computationally intractable.

Therefore, assuming choices of $c, \delta$ which satisfy **(B1)** and **(B2)**, we have shown Cascade is completely secure for any CompNode only executing Layer 0. In practice, these assumptions may not always hold. In fact, learning-based attacks attempt to violate the independence assumption of **(B1)**, through token-infilling between gaps of a $(c, \delta)$-sequence. Furthermore, even as the subset sum problem is NP-complete, this does not mean that the particular choice of sets in our setup are not susceptible to, say, an approximation algorithm. To truly test **(B2)**, future work should explicitly enumerate the linear program and test the efficacy of state-of-the-art solvers on it, perhaps with token-infilling priors integrated. Nevertheless, the above explicit reduction shows that for large enough $c, \delta$, it is quite likely that **(B1)** and **(B2)** nearly hold.

## Q. Cost Optimizations

In Section 7.1, we gave a high-level overview of Cascade, and deferred discussions about optimization. Here, we discuss a few cost and communication optimizations, again assuming symmetry of $S$ and $T$ sharding.

**Caching**    After a new token is generated in Cascade, the CompNode holding that token will send it back to *one* of the existing CompNodes, and single-token generation will repeat to get the next token. To speed up generation after the first new token, the CompNodes and AttnNodes can store their partial intermediate states, and only the 1 CompNode and $\beta$ AttnNodes associated with the most recent token will need to participate in the single-token generation: this means KV-caching naturally extends to Cascade. Formally, suppose $n$ is the index of the most recently generated token, and it belongs to the hidden shard $R_i$ and the query shard $S_j$. Only CompNode$_i$ needs to perform new computation in generating the $(n+1)$st token, along with each AttnNode$_{jk}$ for $1 \le k \le \beta$: this is because only these AttnNodes require the $n$th query row. Furthermore, these $\beta + 1$ nodes, having stored intermediate results from previous forward passes, can avoid repeat computation of attention scores and earlier hidden states. Essentially, this results in token-sharded KV-caching.

**Symmetry Reduction**    We see that AttnNode$_{jk}$ and AttnNode$_{kj}$ actually have the exact same information in the worst-case: they both have access to indices $S_j \cup S_k$. Thus, at no loss of security, we can combine AttnNode$_{jk}$ and AttnNode$_{kj}$ into one node, thereby approximately halving the number of AttnNodes required, and reducing communication byte overhead.

**Synchronization**    A key assumption in our communication analysis from Appendix O was that nodes synchronize between stages. That is, AttnNodes wait until they all finish before sending information to CompNodes in parallel; and likewise for the CompNodes sending information to AttnNodes. But in practice, depending on the sharding scheme, synchronization is not necessary; and relaxing it can allow some nodes to proceed earlier than others. For instance, in a sharding scheme where CompNode$_1$ holds only the first $k$ tokens, because the first $k$ logits do not depend at all on tokens $k + 1, \ldots, n$ in a unidirectional model, then CompNode$_1$ can proceed through all its forward passes without waiting for *any* information from other nodes. Future work could analyze the tradeoff between such synchronization relaxations, which are not possible with schemes like $(c, \delta)$-sharding, and token security.

## R. Cascade Scalability

Here, we demonstrate how Cascade for $\alpha = 2$ without $m$-splitting scales across different model sizes. The below numbers are obtained over 100 runs. Observe that the runtime grows sublinearly with the model size, suggesting Cascade can also scale well to larger models like Llama-2-70B. Furthermore, we emphasize that our baseline Puma on Llama-2-7B (Dong et al., 2023b) took around 300 seconds for a full forward pass, so our approach is over $20\times$ faster.

| Model Name | Model Size (Parameters) | Mean Runtime (s) | 95% Confidence Interval (s) |
| --- | --- | --- | --- |
| Bert-Base | 110M | 0.7 | [0.62,0.74] |
| Bert-Large | 335M | 1.3 | [1.24,1.46] |
| Llama-3.2-1B-Instruct | 1B | 2.6 | [2.33,2.96] |
| Llama-2-7B | 7B | 12.7 | [11.07,14.07] |
| Llama-2-13B | 13B | 22.7 | [20.58,25.99] |

*Table 17.* Model-size-scaling analysis for Cascade. Runtimes and confidence intervals obtained over 100 runs.

# S. Noising method performance

Below, we provide exact (not only the maximum) ROUGE-L scores across layers $1, 6, 11, 16, 21, 26$, for all methods of noising discussed in Section 6. Table 18, Table 19, Table 20 show these results. We also provide a complete breakdown of LiveBench scores per category in Table 22.

*Table 18.* The ROUGE-L scores of decoded texts with added noise and no permutation.

| **Layer** | $\sigma = 10^{-2}$ | $\sigma = 10^{-1}$ | Random Emb | 8-bit quantization | 4-bit quantization |
|---|---|---|---|---|---|
| 1 | 0.9263 | 0.9177 | 0.9309 | 0.8901 | 0.8844 |
| 6 | 0.9273 | 0.3271 | 0.9340 | 0.8726 | 0.8652 |
| 11 | 0.9070 | 0.0856 | 0.8170 | 0.8943 | 0.8764 |
| 16 | 0.9175 | 0.0587 | 0.7552 | 0.8620 | 0.8669 |
| 21 | 0.9232 | 0.0977 | 0.8247 | 0.8834 | 0.8839 |
| 26 | 0.9070 | 0.0485 | 0.6257 | 0.8751 | 0.8771 |

*Table 19.* The ROUGE-L scores of decoded texts with added noise and sequence dimension permutation.

| **Layer** | $\sigma = 10^{-2}$ | $\sigma = 10^{-1}$ | Random Emb | 8-bit quantization | 4-bit quantization |
|---|---|---|---|---|---|
| 1 | 0.0696 | 0.0000 | 0.1683 | 0.8167 | 0.8157 |
| 6 | 0.0354 | 0.0000 | 0.0418 | 0.8236 | 0.8409 |
| 11 | 0.0278 | 0.0011 | 0.0337 | 0.8479 | 0.8138 |
| 16 | 0.0133 | 0.0023 | 0.0202 | 0.8568 | 0.8116 |
| 21 | 0.0136 | 0.0051 | 0.0321 | 0.8283 | 0.8250 |
| 26 | 0.0096 | 0.0096 | 0.0236 | 0.8236 | 0.7956 |

*Table 20.* The ROUGE-L scores of decoded texts with added noise and hidden dimension permutation.

| **Layer** | $\sigma = 10^{-2}$ | $\sigma = 10^{-1}$ | Random Emb | 8-bit quantization | 4-bit quantization |
|---|---|---|---|---|---|
| 1 | 0.0669 | 0.0000 | 0.1945 | 0.7544 | 0.7497 |
| 6 | 0.0353 | 0.0000 | 0.0359 | 0.6696 | 0.6420 |
| 11 | 0.0301 | 0.0009 | 0.0300 | 0.6667 | 0.8138 |
| 16 | 0.0166 | 0.0018 | 0.0144 | 0.6325 | 0.8116 |
| 21 | 0.0164 | 0.0036 | 0.0153 | 0.5029 | 0.8250 |
| 26 | 0.0116 | 0.0101 | 0.0114 | 0.3848 | 0.7956 |

*Table 21.* The ROUGE-L scores of decoded texts with added noise and factorized-2D permutation.

| **Layer** | $\sigma = 10^{-2}$ | $\sigma = 10^{-1}$ | Random Emb | 8-bit quantization | 4-bit quantization |
|---|---|---|---|---|---|
| 1 | 0.0675 | 0.0000 | 0.1919 | 0.7328 | 0.7146 |
| 6 | 0.0346 | 0.0000 | 0.0361 | 0.6075 | 0.5820 |
| 11 | 0.0273 | 0.0016 | 0.0297 | 0.4916 | 0.5753 |
| 16 | 0.0182 | 0.0027 | 0.0140 | 0.3731 | 0.5701 |
| 21 | 0.0196 | 0.0044 | 0.0151 | 0.3845 | 0.5568 |
| 26 | 0.0116 | 0.0120 | 0.0117 | 0.3496 | 0.5564 |

*Table 22.* Performance of Gemma-2-2B-IT on LiveBench with added noise.

| Method | Avg. | Coding | Data Analysis | Instruction Following | Language | Math | Reasoning |
|---|---|---|---|---|---|---|---|
| Baseline (no noise) | 20.7 | 9.4 | 26.1 | 48.9 | 15.2 | 13.1 | 11.3 |
| Gaussian, $\sigma = 10^{-2}$ | 21.0 | 11.1 | 27.4 | 51.2 | 13.7 | 13.4 | 9.3 |
| Gaussian, $\sigma = 10^{-1}$ | 1.2 | 0.0 | 0.0 | 6.9 | 0.4 | 0.0 | 0.0 |
| Random emb. prefix | 21.3 | 8.8 | 27.5 | 50.1 | 16.1 | 13.6 | 12.0 |
| 8-bit quantization | 20.2 | 8.8 | 27.1 | 49.2 | 13.3 | 13.0 | 10.0 |
| 4-bit quantization | 19.1 | 6.5 | 25.5 | 50.5 | 9.5 | 10.9 | 12.0 |

