# OpenReview forum: "Hidden No More: Attacking and Defending Private Third-Party LLM Inference"
_ICML.cc/2025/Conference — ICML 2025 poster_

### Official Review · Reviewer_nbMt · 2025-03-07

**Overall Recommendation:** 2

**Summary:**

The paper investigates the vulnerabilities of private inference in large language models (LLMs). Recent organizations rely on third-party LLM inference services when deploying large models locally due to resource constraints. These setups raise significant privacy concerns as user prompts will be disclosed to untrusted parties.

The authors introduce a white-box reconstruction attack called the vocabulary-matching attack (VMA) that can recover user input text from the hidden states of LLMs with high accuracy. The attack leverages the causal ordering of decoder-only transformers and the finite set of tokens in the vocabulary, iteratively guessing the input token by comparing intermediate hidden states.

To mitigate the vulnerability, the paper introduces Cascade, a multi-party inference scheme based on token-level sharding. Cascade's key idea is to shard hidden states along the token dimension across multiple parties. In this way, no single party can get enough information to reconstruct the user's input. The authors demonstrate that Cascade effectively defends against both their VMA and the existing attack.

## update after rebuttal
Thank the authors for the detailed response. My concern about the white-box access has been addressed. I will raise my rating for the point.

**Claims And Evidence:**

Not always.

**Essential References Not Discussed:**

No.

**Experimental Designs Or Analyses:**

Yes.

**Methods And Evaluation Criteria:**

Could be improved.

**Other Comments Or Suggestions:**

No.

**Other Strengths And Weaknesses:**

The paper has natural weaknesses, which may limit its contributions. First, the attack assumes open-weight access, which may be less applicable in proprietary or black-box LLM inference setups. Besides, if white-box LLM access is granted, the adversary can get the input and output directly. There is no need to mount the proposed attack. Therefore, the VMA seems artificial.

Second, while Cascade prevents the VMA and the existing attack, it does not provide cryptographically rigorous guarantees like SMPC. The authors note that Cascade does not secure individual input embeddings and suggest SMPC for this layer. However, embeddings protected by SMPC provide computational indistinguishability, which means that VMA cannot work in this case.

Third, Cascade's security relies on multi-party setups, which is impractical in typical LLM deployment environments, especially when scale shards are needed. Besides, the choice of sharding parameters (e.g., c and δ) needs careful tuning in practice, which may also challenge Cascade's practical applications.

**Questions For Authors:**

No.

**Relation To Broader Scientific Literature:**

Not sure.

**Theoretical Claims:**

Not applicable.

---

> ### Author Rebuttal · Authors · 2025-03-31
>
> Thank you for your review and for recognizing Cascade's effectiveness against the vocab-matching attack (VMA). We address your comments below.
>
> **W1**
>
> Regarding white-box access: we clarify this differs from open-weights. Our setting involves access to permuted hidden states and weights, but not the corresponding input tokens. We emphasize that the security of schemes [[1]](https://arxiv.org/abs/2405.18744), [[2]](https://arxiv.org/abs/2312.00025) and [[3]](https://arxiv.org/abs/2412.10652) all assume the difficulty of this reversal problem for security. To the best of our knowledge, there is no prior work that demonstrates the insecurity of permuted hidden states.
>
> Furthermore, our attack extends to break the schemes of [2, 3] even in the closed-weights setting. These protocols give permuted model weights and embeddings to the party doing inference, where the embeddings are computed locally by the user. Therefore, the VMA may be applied exactly as mentioned in our submission - except that now the embedding matrix is not known, so the first step of inference in the attack seemingly cannot be done.
>
> However, it is straightforward to bypass this. The adversary can collect the (finite) set of input embeddings in the vocabulary over repeated inference calls, and can perform the VMA by iterating through this set. Decoding of the embeddings to tokens is also possible even if the tokenizer remains private – this essentially constitutes a simple substitution cipher, where each token in the vocabulary is substituted by its embedding vector. This may be easily broken by collecting data over many queries and using straightforward methods such as frequency analysis and contextual modelling.
>
> Therefore, our result extends to breaking the proposed schemes of [2] and [3] even in the closed-weight setting. We further note that the proposed scheme of [1] explicitly provides the model weights to the party performing inference, and so corresponds to the open-weights setting.
>
> **W2**
>
> We wish to clarify your statement that: “The authors note that Cascade does not secure individual input embeddings and suggest SMPC for this layer. However, embeddings protected by SMPC provide computational indistinguishability, which means that VMA cannot work in this case.”
>
> First we assume that you mean that Cascade cannot work in this case. We clarify that here we mean the using SMPC for the first layer, but then _decoding into plaintext_ after it -- e.g. in additive secret sharing, parties send shards of their additive shares to Cascade nodes, who sum them. In this way, Cascade can be used on the plaintext hiddens from layer 1 onwards. Similarly, SMPC can be applied to any number of initial layers, followed by Cascade on the plaintext token-sharded hiddens for the remaining layers.
>
> Regarding lack of cryptographic guarantees - we are clear in our submission that Cascade is _not_ a cryptographic scheme (e.g. lines 80, 321, 414). However, it is our belief that novel defensive methods are of value to the wider research community if they provide sufficient practical defence, even if they do not have formal guarantees that can be proven. We point to the extensive literature on adversarial attacks and defences of neural networks such as [[4]](https://arxiv.org/abs/2406.05927), [[5]](https://arxiv.org/abs/2302.04638), [[6]](https://arxiv.org/abs/2404.09349), none of which have formal guarantees, yet are used commonly in practice due to their efficacy (e.g. see https://robustbench.github.io/).
>
> **W3**
>
> Regarding the tuning of parameters, we have shown from our experiments that for sufficiently large values of $c$ and $\delta$, good security is obtained. Although _optimum_ performance may indeed be dependent on the use case, we are confident that users may follow our prescribed heuristics of $c \geq 8, \alpha \geq 12$ and $m \geq 4$ and achieve good security in the majority of cases.
>
> Regarding the difficulty of obtaining enough nodes, we point to the success of previous projects in decentralized training and inference ([[7]](https://github.com/learning-at-home/hivemind), [[8]](https://arxiv.org/abs/2209.01188), [[9]](https://github.com/PrimeIntellect-ai/prime)) that received contributions from thousands of distinct participants. As such, we are optimistic that sufficient nodes can be gathered in such a setting in order to enable the use of Cascade.
>
> **Conclusion**
>
> Thank you again for your considered review. We have responded to your points above, where in some cases - such as the open-weights setting - we have shown extensibility of our method to also cover the closed-weights setting, and we have also clarified some misunderstandings, such as the compatibility of SMPC with Cascade. We strongly believe in the value of our work, both to show the insecurity of existing proposed schemes, as well as to offer a potential solution to their insecurity. We would be grateful if you would consider raising your score in light of the above.

---

### Official Review · Reviewer_7zaS · 2025-03-14

**Overall Recommendation:** 3

**Summary:**

This paper investigates privacy vulnerabilities in third-party LLM inference services, focusing on an open-weight setting. The authors first propose a vocabulary-matching attack, which can recover original user prompts from intermediate hidden states with near-perfect accuracy and remains effective against various permutation-based and noise-based defenses. Then, a multi-party inference scheme that shards hidden states at the token level is proposed, which is robust against the vocabulary-matching attack while maintaining computational and communication efficiency. Experiments on Gemma-2-2B-IT and Llama-3.1-8B-Instruct validate the effectiveness of the attack and defense.

## Update after rebuttal
Some of my concerns have been addressed, but the paper needs a major revamp in organization and writing. I will keep my score.

**Claims And Evidence:**

Yes.

**Essential References Not Discussed:**

The literature review is comprehensive enough in my opinion.

**Experimental Designs Or Analyses:**

Yes. The attack evaluation methodology is well-structured, covering different LLM architectures, defenses, and layers, while the security of Cascade is evaluated through $(c,\delta)$-sharding experiments.

**Methods And Evaluation Criteria:**

Yes.

**Other Comments Or Suggestions:**

There is a typo in Algorithm 2, line 10.

**Other Strengths And Weaknesses:**

**Strengths:**
1. The attack method is simple yet highly effective, even against some commonly considered defenses based on permutation and perturbation.
2. Extensive experimental results are presented to validate the effectiveness of both the proposed attack and defense.

**Weaknesses:**
1. The paper is quite difficult to follow, the organization and writing need significant improvement. For example, Figure 1 is presented but not explicitly mentioned in the main text. The key steps of the proposed multi-party inference scheme (i.e., Algorithm 2) should be discussed in detail.
2. The proposed defending mechanism, Cascade, lacks formal security guarantees and relies on empirical evaluation.
3. It seems that the attack/defense suffers from high computational and communication overhead, and it is not clear how it works when scaled to larger models (e.g., Llama-3-70B).

**Questions For Authors:**

See my comments about weaknesses.

In addition, it would be helpful if the author could provide some exemplary scenarios in practice, in which a user requires multiple third parties to perform LLM inference as this may incur privacy concerns.

**Relation To Broader Scientific Literature:**

1. The paper builds on prior works on LLM inversion attacks (e.g., Wan et al. (2024), Morris et al. (2023b)), improving attack effectiveness.
2. It challenges the security assumptions of prior permutation-based defenses (Zheng et al. (2024), Yuan et al. (2024), Luo et al. (2024)).
3. The proposed defending mechanism presents a new trade-off between privacy and efficiency, improving over SMPC approaches (Li et al., 2023; Dong et al., 2023b).

**Theoretical Claims:**

This paper is mainly empirical, and not many theoretical results are involved.

---

> ### Author Rebuttal · Authors · 2025-03-31
>
> Thank you for your review. We are glad that you found our attack effective, and found our experimental work to be extensive. We address your comments below.
>
> **W3**
>
> Thank you for raising this important point. We have now run scaling experiments to investigate the effect of model size on the performance of both our attack as well as the defense. We present this below.
>
> | Model Size (num parameters) | Average Attack Time (s) | Vocabulary Size | Model Name              |
> |----------------|--------------------------|------------------|--------------------------|
> | 1B              | 49                     | 128256           | Llama-3.2-1B-Instruct    |
> | 2B              | 124                      | 256000           | Gemma-2-2B-IT            |
> | 8B              | 69                       | 128256           | Llama-3.1-8B-Instruct    |
> | 27B              | 304                       | 256000           | Gemma-2-27B-IT ($\epsilon = 30$)   |
> | 27B              | 124                       | 256000           | Gemma-2-27B-IT ($\epsilon = 40$)   |
>
> This table shows the average attack time over 10 decodings for each of the above models, including the optimizations mentioned in Section 4.2.2 of our submission. As can be seen, the attack time does not significantly increase due to model size, but is primarily a function of the vocabulary size, as well as the parameter $\epsilon$. Even if $\epsilon$ is not very well chosen, the attack still takes on the order of minutes for perfect decoding of length 100 prompts.
>
> Next, we present model-size-scaling analysis for Cascade:
>
> | Model Size (num parameters) | Mean Runtime (s) | 95% Confidence Interval (s) |
> |--------------------------|------------------|------------------------------|
> | 110M                    | 0.7            | 0.62 – 0.74              |
> | 335M                    | 1.3            | 1.24 – 1.46              |
> | 1B                      | 2.6            | 2.33 – 2.96              |
> | 7B                      | 12.7           | 11.07 – 14.07            |
> | 13B                     | 22.7           | 20.58 – 25.99            |
>
> The above numbers are obtained over 100 runs, with $\alpha = 2$. The models for the 110M and 335M sizes are BERT-base and BERT-Large, and for 1B-13B are Llama 2. We see that the runtime grows sublinearly with the number of parameters. We emphasize that our baseline, [[1]](https://arxiv.org/abs/2307.12533), reported a single forward pass on Llama 7B in approximately 300s, so our approach is ~24x faster. Due to computational constraints, we have not yet run on 70B, but we believe the same scaling law will apply as above. We have now updated our local revision to include the above results, and will include those - as well as the 70B result - in our camera ready version.
>
> **W2**
>
> Thank you for mentioning this point. We agree, and we are very clear in our submission that Cascade is not a cryptographic scheme (e.g. lines 80, 321, 414). However, it is our belief that novel defensive methods are of value to the wider research community if they provide sufficient practical defence, even if they do not have formal guarantees that can be proven. We point to the extensive literature on adversarial attacks and defences of neural networks such as [[2]](https://arxiv.org/abs/2406.05927), [[3]](https://arxiv.org/abs/2302.04638), [[4]](https://arxiv.org/abs/2404.09349), none of which have formal guarantees, yet are used commonly in practice due to their efficacy (e.g. see https://robustbench.github.io/).
>
> **Q2**
>
> To be clear, the use of multiple parties in Cascade is to provide additional security via token-sharding. For feasibility of obtaining enough parties, we point to the success of previous projects in decentralized training and inference ([[5]](https://github.com/learning-at-home/hivemind), [[6]](https://arxiv.org/abs/2209.01188), [[7]](https://github.com/PrimeIntellect-ai/prime)) that received contributions from thousands of distinct participants over diverse geographies. As such, we are optimistic that sufficient nodes can be gathered in such a setting in order to provide good security.
>
> **W1**
>
> We agree with all the points you have raised here. We have now included references to Figure 1 in the text, reorganized certain sections such as experimental results and related works, and brought some of the Appendix D details regarding Algorithm 2 to the main body. We also thank you for your eagle-eyed spotting of the typo in Algorithm 2, which we have now fixed.
>
> Thank you once again for your thoughtful and considered review. We strongly believe in the value of our work both to highlight the security inadequacy of permuted hidden states proposed by several existing protocols, and our proposed solution as a potential method to address it. In light of our addressal of your points above, we would be very grateful if you would consider raising your score. Thank you!

---

### Official Review · Reviewer_RAtg · 2025-03-14

**Overall Recommendation:** 2

**Summary:**

This manuscript explores the field of private inference and proposes a vocabulary-matching attack that exploits hidden states to recover the original input of an LLM. The authors highlight that existing permutation-based and noise-based schemes fail to provide sufficient security against such an attack. To address this vulnerability, the manuscript introduces a token-sharded multi-party inference framework that is crypto-free. Experimental results demonstrate its performance improvements over existing cryptographic methods.

**Claims And Evidence:**

Partially.
See Sec. Other Strengths And Weaknesses for detailed comments.

**Essential References Not Discussed:**

The reviewer recommends that the authors consider [1], as it relates to the security of the proposed method.
See Sec. Other Strengths And Weaknesses for detailed comments.
[1] Wong, Harry WH, Jack PK Ma, Donald PH Wong, Lucien KL Ng, and Sherman SM Chow. "Learning Model with Error-Exposing the Hidden Model of BAYHENN." In IJCAI, pp. 3529-3535. 2020.

**Experimental Designs Or Analyses:**

Experiments were conducted to support the manuscript's performance claims.
However, some experiments are not properly designed or require further clarification.
See Sec. Other Strengths And Weaknesses for detailed comments.

**Methods And Evaluation Criteria:**

The proposed methods have limited practicality and potential security concerns.
See Sec. Other Strengths And Weaknesses for detailed comments.

**Other Comments Or Suggestions:**

N/A.

**Other Strengths And Weaknesses:**

This paper introduces an attack method against permutation-based and noise-based defenses while proposing a crypto-free scheme to prevent original input reconstruction from intermediate hidden states.

### Strengths:
+ The manuscript effectively highlights the performance limitations of crypto-based schemes, which hinder their adoption in real-world applications.
+ It identifies potential security risks in existing defense methods.
+ The manuscript is comprehensive, and its experimental results are notable.

### Weaknesses:
However, the reviewer has identified several key issues, which lead to the recommendation to reject the manuscript in its current form.

#### 1. Insufficient Technical Contribution
- While the proposed vocab-matching attack successfully breaks existing permutation-based and noise-based defenses, it primarily relies on an approximation-based brute-force search method. The attack strategy minimizes the distance between current and original input states, which is a straightforward extension of existing methods.
- The proposed defense method is fundamentally based on splitting the hidden states into multiple segments, ensuring no single entity has access to a complete hidden state. However, the idea is intuitive and lacks sufficient technical depth for publication in ICML.

#### 2. Security Concerns and Practical Limitations
- According to [1], exposing intermediate plaintext values (rather than encrypted values) can still lead to privacy leakage. This concern needs to be formally addressed with rigorous proofs.
- While the scheme prevents full exposure of consecutive values in the hidden state vector, parties can still access partial consecutive values. The authors must prove whether this partial exposure can aid in reconstructing parts of the user’s input.
- Regarding the SoftMax layer, many variants use the formulation:
  $
  e^{x_i - \max(x)} / \sum_j e^{x_j - \max(x)}
  $
  Under the current sharding technique, how can the max(x) term be computed without cryptographic operations? If the max operation requires inputs from all parties, would this contradict the manuscript's core design principle, which restricts each party’s access to partial data?
- Given these concerns, the current security proof, which claims that search complexity is sufficiently large to ensure security, is inadequate. The authors must provide a more rigorous, formal security proof with numerical analysis, rather than relying on limited empirical experiments with specific attacks and models.

#### 3. Issues in Presentation and Writing Quality
- The manuscript’s writing is informal, requiring substantial improvement.
- The notation system is inconsistent and incomplete. The authors should provide a comprehensive notation table, rather than defining symbols only when they first appear.
- Avoid using coding-style expressions such as `a[:, R_i, T_k]` without explicit explanation. While these may be common in Python-based implementations, they are not standard in academic writing.
- The organization of the manuscript needs restructuring. While space constraints exist, too many crucial details, including the construction of the inference protocol, are placed in the appendix. It is recommended to integrate "Existing Work" and "Related Work" into a single section for better readability.
- Clarify the conceptual rationale behind the proposed method.
  - Figure 1 should be redesigned to better illustrate the scheme. The current version is difficult to interpret unless the reader has prior knowledge of the method.

#### 4. Experimental Design Issues
- While the reviewer acknowledges the significant overhead of MPC-based methods, the manuscript fails to accurately compare the performance gap between crypto-based methods and the proposed approach.
- The experimental setup does not properly account for potential optimizations in cryptographic schemes:
  - The manuscript assumes model weights are public while user input is private.
  - With this assumption, many expensive MPC operations can be replaced by more efficient alternatives, such as:
    - Secure multiplication via Homomorphic Encryption (HE)
    - Beaver’s triplets for multiplication
  - However, the baseline cryptographic methods (PUMA, MPCFormer) were designed for settings where both weights and inputs are private, leading to an unfair comparison.
- Unless the authors can explicitly clarify and adjust the baselines to match the same setting, the current performance claims remain questionable.

[1] Wong, Harry WH, Jack PK Ma, Donald PH Wong, Lucien KL Ng, and Sherman SM Chow. "Learning Model with Error-Exposing the Hidden Model of BAYHENN." In IJCAI, pp. 3529-3535. 2020.

**Questions For Authors:**

N/A.

**Relation To Broader Scientific Literature:**

Despite its shortcomings, this manuscript introduces the potential for crypto-free or lightweight crypto-based techniques in developing more efficient privacy-preserving schemes.

**Theoretical Claims:**

The proofs have been checked.
See Sec. Other Strengths And Weaknesses for detailed comments.

---

> ### Author Rebuttal · Authors · 2025-04-01
>
> Thank you for your detailed review of our submission. We appreciate that you found our paper effectively identifies potential security risks in existing privacy-preserving schemes and that our work is comprehensive in coverage.
>
> **W4**
>
> We agree with your point. We have now modified the implementation of MPCFormer in Crypten ([[1]](https://arxiv.org/abs/2109.00984)), the SMPC framework they used, to support the use of public weights. Crypten uses Beaver’s triples for matrix multiplication - see Appendix C.1.1 of their paper. For fair comparison, we run on the same setup as Cascade. The table below shows means and 95% confidence intervals over 100 runs:
>
> | **Scheme**                        | **BERT-Base Runtime (s)**        | **BERT-Large Runtime (s)**   |
> |----------------------------------|----------------------------------|----------------------------------|
> | MPCFormer (private weights) | 339.35 [311.58, 396.92] | 1407.26 [1281.12, 1680.31] |
> | MPCFormer (public weights) | 49.40 [45.61, 57.29]     | 143.88 [131.00, 178.64] |
> | Cascade$_{α=2}$         | 0.66 [0.615, 0.738]                  |  1.33 [1.24, 1.46]    |
> | Cascade$_{α=4}$              | 0.59 [0.51, 0.69]             | 1.57 [1.44, 1.73]             |
> | Cascade$_{α=8}$              | 0.74 [0.62, 0.96]             | 1.58 [1.27, 1.97]             |
>
> MPCFormer indeed benefits from a significant speedup by the use of public weights in its protocol - approximately in the range of 7-10x. However, this is still 50-100x slower than Cascade. Moreover we note that the runtime of Cascade does not seem to grow very quickly as a function of $\alpha$.
>
> We will also update the result of PUMA. This requires the use of a separate SMPC framework ([[2]](https://github.com/secretflow/spu)). We will have the experimental results for this ready for the final draft of the paper.
>
> **W2**
>
> Thank you for bringing to our attention the work of [[3]](https://www.ijcai.org/proceedings/2020/488). It is a stark reminder of the importance of applying care when making claims about the strength of guarantees provided by a scheme – as with the claimed strength of plaintext permuted hidden states, which we demonstrate are easily decodable. We note that the work [3] devises a successful attack against, BAYHENN, claimed the same formal guarantees of privacy as FHE, by misapplication of the LWE assumption. However we are clear in our submission that Cascade is _not_ a cryptographic scheme (e.g. lines 80, 321, 414), and therefore do not claim any formal guarantees against e.g. learning-based attacks.
>
> The subtract-max variant of softmax works by calculating the maximum $m^x_k$ over each partial row $a^x_k$ of the attention scores given to AttnNodes, where $x$ is the row and $k$ is the column index of the AttnNode – as well as $v^x_k=expsum(a_k^x-m_k^x)$. Thus in standard softmax, CompNode$_i$ gets $o^x_k=softmax(a^x_k)V_k,w^x_k=expsum(a^x_k)$ for all $k$ and rows $x\in R_i$; and for subtract-max, it gets $o^x_k,m^x_k,v^x_k$.
>
> We analyze the security of standard softmax in Appendix J, showing that it necessitates large ‘gaps’, which $(c,\delta)$ sharding satisfies. We can now also show such gaps are _sufficient_ for first layer security – by showing equivalence to the subset-sum problem over vectors. Due to the 5000 character limit, we will include details of this in our next response. We briefly mention here that the subtract-max variant simply adds 1 to the value of $c$ required for security over standard softmax.
>
> **W1**
>
> Our attack is not brute-force; it exploits the properties of attention and a finite dictionary to break permuted hidden states in linear time rather than the exponential time that is assumed in the schemes of [[4]](https://arxiv.org/abs/2405.18744), [[5]](https://arxiv.org/abs/2312.00025) and [[6]](https://arxiv.org/abs/2412.10652). [6] even has a ‘proof’ using a misapplication of distance correlation theory, that is incorrect; we will include further details in the next response. The attack also requires suitable ‘matching functions’ such as sorted-L1-distance (see lines 234-235). Further, practical success of this attack was not obvious a-priori, due to the possibility of many ‘collisions’ of states when matching over all N hidden states - the result of nearly 0 collisions is intriguing in its own right.
>
> Moreover, our proposal for Cascade has not to our knowledge been proposed in prior literature. Even if it is considered intuitive, our extensive experimental coverage of performance and security is, we think, of value to the wider community.
>
> **W3**
>
> Thank you for your points. We agree with your suggestions, and have added a notation table and modified the Python-style notation.
>
> Thank you once again for your thorough feedback. Even if we do not fully align on the value of our work, we feel that your review was particularly thoughtful and of high value. We look forward to following up with our next response to expand on some of the above points in further detail.

---

### Official Review · Reviewer_dxey · 2025-03-14

**Overall Recommendation:** 2

**Summary:**

This paper proposes a vocabulary matching attack by exploiting the autoregressive characteristics of the generative model, which can attack the privacy-preserving large language model (LLM) inference framework based on permutation and noise under the assumption that the model parameters are public. At the same time, the author introduces a sharding-based privacy-preserving LLM inference framework Cascade. Cascade uses the token sharding method in the sequence dimension to maintain computational and communication efficiency, while providing security against the proposed attack and previous reversal methods.

## update after rebuttal
I thank the authors for their detailed responses. After reading the responses and comments from other reviewers, I think this paper requires major revisions regarding the writing, the theoretical security discussion, and the practicality of the multi-party setup. Therefore, I kept my score.

**Claims And Evidence:**

Overall, the claims made in this paper are supported by clear and convincing evidence but there are still areas for improvement. In particular,

1. Using only the ROUGE score to measure the security of Cascade is not sufficient. More indicators and evidence are needed to enhance it.

2. Considering only the two attack methods of vocabulary matching and learning is not comprehensive. Experimental results of other attack methods are needed.

**Essential References Not Discussed:**

No.

**Experimental Designs Or Analyses:**

The experimental designs and analyses in the paper are generally sound and valid, particularly for the vocabulary-matching attack. The evaluation of the Cascade defense mechanism is reasonable but could be strengthened with more direct security assessments and additional experimental validations.

**Methods And Evaluation Criteria:**

The proposed methods and evaluation criteria are well-aligned with the problem of privacy-preserving LLM inference. They address both the attack and defense aspects comprehensively using appropriate datasets and metrics. Some additional evaluations could further strengthen the conclusions, but overall, the methods and criteria make sense for the problem at hand.

**Other Comments Or Suggestions:**

1. The notation becomes quite dense in sections describing Cascade's implementation. A notation table would help readers keep track of variables and sharding parameters.

2. The algorithm description (Algorithm 1) could benefit from a step-by-step example to illustrate how it works in practice.

3. The security analysis (Section 7.2) would be clearer with a diagram showing how token sharding prevents reconstruction attacks.

**Other Strengths And Weaknesses:**

**Strengths:**

1. This paper proposes an effective attack method for the privacy-preserving LLM inference framework with random permutation and noise mechanism in the scenario where the model parameters are public.

2. This paper proposes a privacy-preserving LLM inference algorithm based on token sharding, which is new to the area.

**Weaknesses:**

1. The effectiveness of the word list matching attack is mainly reflected in the open-weight setting, which may not represent all deployment scenarios. In closed-weight or more restricted environments, the word list matching attack will fail.

2. The actual deployment of Cascade in a distributed environment may face challenges that are not fully addressed in this paper, such as network reliability and synchronization issues.

3. In terms of security, Cascade cannot achieve theoretical security, and the security strength is related to the number of nodes. Finding a large number of non-colluding nodes in actual deployments seems difficult to achieve.

**Questions For Authors:**

1. The attack assumes access to the full model architecture and weights. How would the effectiveness of the attack change in scenarios where the model architecture or weights are not fully known to the adversary? Would the attack still be feasible with partial knowledge?

2. The security analysis assumes certain sharding parameters $(c, \delta)$. How would the security properties change if an adversary could influence or discover these parameters in a real-world deployment?

**Relation To Broader Scientific Literature:**

The paper makes contributions by both advancing attack methodologies and proposing a practical defense mechanism that addresses demonstrated vulnerabilities, filling gaps in the literature on privacy-preserving LLM inference.

**Theoretical Claims:**

The security proof in Section 7.2 of the paper is reasonable, but requires more formal treatment, e.g., supplementary cryptographic security analysis.

---

> ### Author Rebuttal · Authors · 2025-03-31
>
> Thank you for your detailed review. We are glad that you found our proposed attack to be effective in the open-weights setting, and that Cascade is novel. We provide responses to some of the points you have raised below.
>
> **W1 + Q1**
>
> A slight extension of our vocab-matching attack can additionally break the schemes of [[1]](https://arxiv.org/abs/2312.00025) and [[2]](https://arxiv.org/abs/2412.10652) in the closed-weights setting. Due to space constraints, please find the details of this in the response to reviewer nbMt.
>
> **W2**
>
> Thank you for raising this important point. We have now tested Cascade in a real life WAN network setting with up to 18 different physical machines, and 72 different logical nodes. Our full results for single layer inference on BERT under these settings is shown below:
>
> | **Scheme**                        | **BERT-Base Runtime (s)**        | **BERT-Large Runtime (s)**       |
> |----------------------------------|----------------------------------|----------------------------------|
> | MPCFormer                        | 55.320                           | 141.222                          |
> | PUMA                             | 33.913                           | 73.720                           |
> | Cascade$_{α=2}$         | 0.662 [0.615, 0.738]                  |  1.331 [1.237, 1.464]    |
> | Cascade$_{α=4}$              | 0.588 [0.513, 0.688]             | 1.572 [1.441, 1.734]             |
> | Cascade$_{α=8}$              | 0.742 [0.622, 0.962]             | 1.584 [1.271, 1.965]             |
> | Vanilla Inference                        | 0.091 [0.084, 0.121]             | 0.273 [0.200, 0.993]             |
>
> The above table shows the average runtimes for Cascade under three choices of $\alpha$. Larger $\alpha$ has more nodes. Mean results are given over 100 runs, and a 95% confidence interval is additionally shown in brackets. Values for MPCFormer and Puma are taken from their respective papers.
>
> We also measured the total communicated bytes in the same setting. Even in the most expensive $\alpha = 8$ setting, Cascade is $\sim150\times$ more efficient in total bytes transferred than the baselines. We conclude that although Cascade will be negatively impacted by poor network conditions, this is true for any SMPC method, and the effect of this will be less deleterious on Cascade than other protocols.
>
> **W3**
>
> We point to the success of previous projects in decentralized training and inference ([[3]](https://github.com/learning-at-home/hivemind), [[4]](https://arxiv.org/abs/2209.01188), [[5]](https://github.com/PrimeIntellect-ai/prime)) that received contributions from thousands of distinct participants over diverse geographies. As such, we are optimistic that sufficient nodes can be gathered in such a setting in order to provide good security.
>
> **Q2**
>
> In fact, our security analysis already assumes the worst case of perfect knowledge of the security parameters $c$ and $\delta$ by the adversary. In practical deployment, the exact sharding scheme may not be known (it can even be changed or randomized continually). We have now made this point more clearly in our local draft and will update it for the camera-ready version.
>
> **Claims 1**
>
> We agree with your assessment. We have now additionally computed BLEU, F1 and token accuracy, as well as provided reconstruction examples in the local draft.
>
> **Claims 2**
>
> Examining the literature, we did not find existing methods of attack that are not learning-based. Are there particular alternative methods that you believe are applicable to this setting?
>
> **Theoretical Claims**
>
> We are clear in our submission that Cascade is _not_ a cryptographic scheme (e.g. lines 80, 321, 414). However, we do examine all possible sources of leakage in Appendix J of our submission.
> Moreover, it is our belief that novel defensive methods are of value to the wider research community if they provide sufficient practical defence, even if they do not have formal guarantees that can be proven. We point to the extensive literature on adversarial attacks and defences of neural networks such as [[6]](https://arxiv.org/abs/2406.05927), [[7]](https://arxiv.org/abs/2302.04638), [[8]](https://arxiv.org/abs/2404.09349), none of which have formal guarantees, yet are used commonly in practice due to their efficacy (e.g. see https://robustbench.github.io/).
>
> **Other Comments**
>
> We agree with all of the readability points you have suggested, and have now included them in our local draft. We will make these changes to the camera-ready version as well.
>
> Thank you once again for your thorough review and your insightful comments. If you have any further comments, or if you think our submission can be improved in any way, please let us know. We have made a significant effort to address each of your points and would appreciate it if you would consider raising your score in light of our response. Thank you!

---

### Decision · Program_Chairs · 2025-05-01

**Decision:**

Accept (poster)

**Comment:**

This work proposes an attack against permutation- and noise-based private inference methods when the weights are public. The attack is innovative and can decode model inputs with high accuracy. Based on this, this work then proposes a heuristic defense that shows success empirically in defending against various forms of leakage.

Overall, reviewers had consensus that this work had lacking clarity and that this work should be revised to improve the presentation of the work and its technical details and that the lacking security guarantees for the defence significantly detract from the efficacy of the approach. Some significant improvements were made during the rebuttal including extensions of the attack to closed-weight settings and more thorough experiments (including a fix of the experimental setting for a proper comparison).

Beyond this, reviewers were also concerned with the technical novelty of this work, as the attacks leverage simple observations to compute a brute-force search. Despite this, the AC believes that these insights have led to an effective attack that warrants merit.